# STEP-BY-STEP VIDEO-TO-AUDIO SYNTHESIS VIA NEGATIVE AUDIO GUIDANCE

## ABSTRACT

We propose a step-by-step video-to-audio (V2A) generation method for finer controllability over the generation process and more realistic audio synthesis. Inspired by traditional Foley workflows, our approach aims to provide better controllability by enabling incremental generation of desired sound, thus enabling users to produce multiple sound events induced by a video comprehensively. To avoid the need for costly multi-reference video–audio datasets, each generation step is formulated as a negatively guided V2A process that discourages duplication of existing sounds. The guidance model is trained by finetuning a pre-trained V2A model on audio pairs from adjacent segments of the same video, allowing training with standard single-reference audiovisual datasets that are easily accessible. Objective and subjective evaluations demonstrate that our method enhances the separability of generated sounds at each step and improves the overall quality of the final composite audio, outperforming existing baselines.

## 1 INTRODUCTION

We are interested in generating realistic audio signals that align seamlessly with given visual contents — a process referred to as Foley (Ament, 2021) in film or game production. In traditional workflows, Foley artists begin with field-recorded or library sounds and incrementally layer in missing elements (e.g., footsteps or fabric movements) to enhance the realism of the audio. While essential for high-quality audiovisual content, this workflow is labor-intensive and time-consuming because even short clips often contain numerous audible events.

Recent video-to-audio (V2A) models (Viertola et al., 2025; Luo et al., 2023; Wang et al., 2024b;a; Liu et al., 2024; Cheng et al., 2025; Polyak et al., 2024a) show promise in automating this workflow. These models produce high-quality audio that semantically and temporally aligns with input videos. However, most models generate an entire track in a single pass and do not offer a mechanism for incremental refinement (i.e., supplementing sounds missing in the generated results). This non-interactive design poses a significant challenge: if the output is missing specific events, creators are compelled to regenerate the entire track. Such inefficiencies limit the practical application of these models, particularly in collaborative workflows with human creators.

We argue that a step-by-step generation mechanism is crucial for practical V2A synthesis (Fig. 1). A model should generate not only a complete track aligned with the video, but also complementary audio that fills missing events without duplicating sounds already existing.[1] This offers greater control and efficiency in the sound creation process, as in the traditional Foley workflow.

A critical challenge to achieve this step-by-step generation is the scarcity of datasets. A straightforward approach (i.e., training a conditional generation model that produces multiple plausible audio tracks per video) requires multi-reference video-audio pairs, which are difficult to obtain at scale. In this paper, we propose a guided generation method, Negative Audio Guidance (NAG), for the step-by-step video-to-audio synthesis without requiring specialized training datasets. We train a model that generates audio semantically similar to the reference audio, using pairs of audio

---

[1]One might consider text-conditional V2A already provides sufficient control for the target audio event to be generated. However, existing text-conditional V2A models struggle to suppress the already generated sound, especially for the prominent event in the video (e.g., in Fig. 4, the moose's footstep sounds are produced in all tracks regardless of the input text prompts).

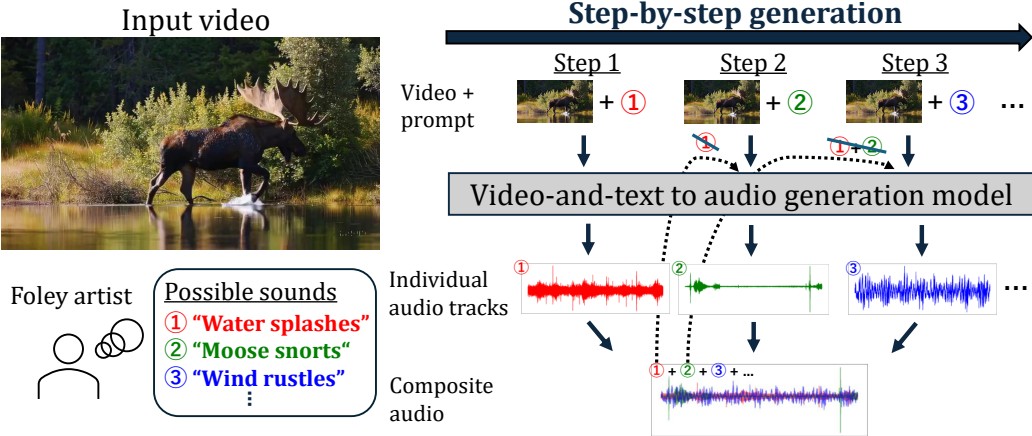

Figure 1: Step-by-step video-to-audio generation for compositional sound effect creation. Video often contains numerous audible events, and Foley artists synthesize composite audio by adding missing audio components step-by-step. Supporting this step-by-step mechanism with a video-to-audio generation model offers greater control and efficiency in the sound creation process.

segments sampled from adjacent segments of the same video. At inference, we utilize the trained model negatively to ensure the generated audio is dissimilar to existing audio, steering generation toward complementary content. By iterating this guided generation process, a model progressively builds a composite audio that covers all relevant sound events in the video. Extensive experiments demonstrate that our method enables step-by-step completion of missing sounds and enhances final audio quality while ensuring the separability of the generated audio at each step.

## 2 RELATED WORK

### 2.1 VIDEO-TO-AUDIO SYNTHESIS

The goal of video-to-audio synthesis is to generate an audio signal that aligns semantically and temporally with an input video. Early approaches used regression models (Chen et al., 2020b) and GANs (Iashin and Rahtu, 2021), while more recent ones have adopted autoregressive models (Viertola et al., 2025) and diffusion models (Luo et al., 2023; Wang et al., 2024b;a; Liu et al., 2024; Polyak et al., 2024a; Chen et al., 2025; Cheng et al., 2025; Wang et al., 2025) due to their high capability in generation tasks. However, these models typically accept only videos (and optionally text prompts) as input conditions, making it impossible to specify sounds that users may want to combine with the generated audio.

Few studies have explored audio conditioning in video-to-audio synthesis to address their respective problem setting. MultiFoley (Chen et al., 2024b) uses conditional audio as a reference for the generated audio. Sketch2Sound (García et al., 2025) takes a similar approach, but only uses a particular set of signal features extracted from the original conditional audio to accept sonic or vocal imitations as conditions. Action2Sound (Chen et al., 2024a) focuses on disentangling foreground and ambient sound, using conditional audio to specify the appearance of ambient sound in the generated audio. ReWaS (Jeong et al., 2025) introduces audio-energy conditioning predicted from video into a text-to-audio generator via ControlNet (Zhang et al., 2023), enabling audio generation that is temporally aligned with the video. Unlike these studies, we utilize audio conditioning to specify what kind of audio *should not* appear in the generated audio, enabling step-by-step generation in video-to-audio synthesis.

### 2.2 GENERATIVE "ADD" OPERATION

Generative "add" operations in the audio domain are executed to generate audio that can be mixed with an input audio signal, often guided by a text prompt. These operations have been explored in

text-to-audio (Wang et al., 2023; Jia et al., 2025) and text-to-music (Han et al., 2024; Parker et al., 2024; Mariani et al., 2024; Postolache et al., 2024; Karchkhadze et al., 2025) synthesis and can be divided into two approaches: training-based and training-free.

In the training-based approach, the model is explicitly trained to perform the "add" operation given the input audio. This training requires a triplet comprising an input audio, a text prompt, and an audio to be added as training data (Wang et al., 2023; Han et al., 2024; Parker et al., 2024). Unfortunately, these methods are difficult to apply in our setting because such data is hard to obtain. Even within a single scene, a mixture of many sounds can be observed, and separating them into individual ones is challenging (Owens and Efros, 2018; Zhu and Rahtu, 2020; Song and Zhang, 2023). SonicVisionLM (Xie et al., 2024) introduces a timestamp-conditioned video-and-text-to-audio model that first converts video into text and then generates audio via a T2A backbone. While this design avoids audio contamination from visual features and enables additive video-to-audio generation, it loses the fine-grained audiovisual cues captured by multimodal V2A models (Polyak et al., 2024a; Chen et al., 2025; Cheng et al., 2025; Wang et al., 2025). As a result, it struggles with subtle or weakly visible events and relies on specialized video–audio–text–timestamp datasets that are costly to build.

On the other hand, the training-free approach is more flexible as it leverages a pre-trained text-to-audio/music model without any specific training process. The "add" operation is conducted as a partial generation of multi-track audio (Mariani et al., 2024; Postolache et al., 2024; Karchkhadze et al., 2025) or a re-generation with a target prompt from structured noise obtained through inverting the input audio (Jia et al., 2025). Instead of specific training data, these methods require particular properties in the pre-trained model: multi-track joint generation (Mariani et al., 2024; Karchkhadze et al., 2025), data-space diffusion models (Postolache et al., 2024), and specific types of model architectures (Jia et al., 2025), which limit their applicability to our video-to-audio setting.

Similar "add" operations have been explored as object insertion in computer vision, where models generate an object image to be added to an input background image. They are also categorized into training-based (Singh et al., 2024; Canberk et al., 2024) and training-free approaches (Tewel et al., 2025), but in either case, they rely on segmentation models to create training data or guide the generation process. It means that a particular subset of pixels in the image is assumed to be wholly replaced with the generated one through the "add" operation. As "add" in the audio domain involves mixing rather than replacing, these approaches cannot be directly applied to our setting.

We adopted a training-based approach for this study, utilizing the "add" operation in the audio domain, and designed our framework to eliminate the need for specialized training data. This approach makes it more practical and adaptable for video-to-audio synthesis tasks.

## 3 METHOD

### 3.1 PRELIMINARIES

**Generative modeling with flow-matching.** Let $p_1(x)$ be a data distribution where $x \in \mathbb{R}^d$. Flow matching (Lipman et al., 2023) considers the probability flow ODE $\frac{d}{dt}\phi_t(x) = u_t(\phi_t(x))$, where $t \in [0, 1]$ is a timestep, $u_t$ is the velocity field, and $\phi_t(x) = \phi(x, t) : \mathbb{R}^d \times \mathbb{R} \to \mathbb{R}^d$ is the flow that maps $x$ to the intermediate data $x_t$. $\phi_t$ can be an arbitrary function that satisfies the terminal condition $\phi_1(x_1) = x_1$ and $\phi_0(x_1) \sim p_0$, where $p_0$ is a tractable distribution such as a standard normal distribution $\mathcal{N}(0, I)$. Following the most popular setting, we define $\phi_t(x_1) = tx_1 + (1-t)x_0$, where $x_0 \sim N(0, I)$, resulting in $u(\phi_t(x)) = x_1 - x_0$.

Solving ODE from $t = 0$ to $1$ with an initial sample $x_0 \sim p_0(x)$ enables sampling from the target data distribution $x_1 \sim p_1(x)$. To achieve this, a neural network is trained to predict $u_t(\phi_t(x))$, which corresponds to $x_1 - x_0$ in our case, by minimizing the squared error over both data and timesteps. In text-conditioned video-to-audio synthesis, the model takes additional conditional inputs, which are input video $V$ and text prompt $C$, to model a conditional flow $u_t(\phi_t(x)|V, C)$.

**Guidance for flow-matching models with multiple conditions.** Classifier-free guidance (Ho and Salimans, 2021) is widely used to improve generation quality and fidelity to conditions. This guidance is typically conducted with a single condition, and it is not trivial to extend it to two conditions,

as in text-conditioned video-to-audio synthesis. To derive a proper guidance process, $p(x|V, C)$ is decomposed as the following equation:

$$p(x|V, C) = p(x)\left(\frac{p(x|V)}{p(x)}\right)\left(\frac{p(x|V, C)}{p(x|V)}\right). \tag{1}$$

On the basis of this decomposition, Kushwaha and Tian (2025) proposed the following guided flow:

$$\tilde{u}_\theta(x_t) = u_\theta(x_t, t, \varnothing, \varnothing) + w_1(u_\theta(x_t, t, V, \varnothing) - u_\theta(x_t, t, \varnothing, \varnothing))$$
$$+ w_2(u_\theta(x_t, t, V, C) - u_\theta(x_t, t, V, \varnothing)), \tag{2}$$

where $\theta$ is a set of the model parameters, and $\varnothing$ denotes a null condition. The three terms of the right-hand side of Eq. (2) respectively correspond to the three factors on the right-hand side of Eq. (1). They empirically show that setting $w_1 = w_2$ achieves better results, and in this case, $u_\theta(x_t, t, V, \varnothing)$ cancels out, which gives us the following simplified formulation:

$$\tilde{u}_\theta(x_t) = u_\theta(x_t, t, \varnothing, \varnothing) + w_1(u_\theta(x_t, t, V, C) - u_\theta(x_t, t, \varnothing, \varnothing)). \tag{3}$$

### 3.2 PROBLEM SETTING: STEP-BY-STEP GENERATION IN VIDEO-TO-AUDIO SYNTHESIS

We are interested in iteratively generating missing audio that complements previously generated audio. Let $x^{(1)}$ be the audio generated at the previous step, $x^{(2)}$ be the target audio to be generated at this step, $C_2$ be the text prompt that specifies the sound event for $x^{(2)}$ missed by $x^{(1)}$, and $V$ be the input video. Our goal is to sample $x^{(2)}$ from $p\big(x^{(2)}\big|V, C_2, x^{(1)}\big)$ so that $x^{(2)}$ corresponds to the concept described by $C_2$, semantically and temporally aligns with $V$, and does not contain duplicated audio present in $x^{(1)}$.

From a straightforward standpoint, learning to generate samples from this distribution would require tuples of $(x^{(2)}, V, C_2, x^{(1)})$ as training data. Unfortunately, constructing such data from a video is a challenging task called visually-guided audio source separation (Owens and Efros, 2018; Zhu and Rahtu, 2020; Song and Zhang, 2023), and thus, we cannot expect high-quality training datasets. Instead, we propose an alternative training framework that eliminates the need for such specialized data.

Note that this processing can be applied to the following generation step without loss of generality. In the $k$-th generation step, we can set the mix of the previously generated $(k-1)$ audio as $x^{(1)}$, and use it to sample $x^{(2)}$. Please refer to Section 5.1 for more details of this procedure.

### 3.3 FORMULATION OF TARGET DISTRIBUTION WITH CONCEPT NEGATION

Recall that we want to generate $x^{(2)}$ to only cover a missing sound event in $x^{(1)}$. In this sense, conditioning by $x^{(1)}$ corresponds to a concept negation (Du et al., 2020; Liu et al., 2022a; Valle et al., 2025) in generating $x^{(2)}$; the generated $x^{(2)}$ *should not* contain any concepts related to $x^{(1)}$. We denote this type of audio condition as $\bar{\mathcal{E}}(\cdot) = \neg\mathcal{E}(\cdot)$ to explicitly differentiate it from a standard type of audio condition denoted by $\mathcal{E}(\cdot)$. Based on the above-mentioned relationship between $x^{(1)}$ and $x^{(2)}$, we approximate the target distribution of $x^{(2)}$ using the concept negation as follows:

$$p\big(x^{(2)}\big|V, C_2, x^{(1)}\big) \approx p\big(x^{(2)}\big|V, C_2, \bar{\mathcal{E}}\big(x^{(1)}\big)\big). \tag{4}$$

Following the study by Du et al. (2020), we assume that the concept negation holds this property:

$$p(x, c_p, \neg c_n) \propto p(x)p(c_p|x)p(c_n|x)^{-1}, \tag{5}$$

where $c_p$ and $c_n$ denote conditional concepts to generate $x$.

To derive our guidance, we decompose the target distribution using Eq. (5) and Bayes' theorem as:

$$p\big(x^{(2)}\big|V, C_2, \bar{\mathcal{E}}\big(x^{(1)}\big)\big) \propto p\big(x^{(2)}, V, C_2, \bar{\mathcal{E}}\big(x^{(1)}\big)\big)$$
$$\propto p\big(x^{(2)}, V\big)p\big(C_2\big|x^{(2)}, V\big)p\big(\mathcal{E}\big(x^{(1)}\big)\big|x^{(2)}, V\big)^{-1}$$
$$\propto p\big(x^{(2)}\big)\left(\frac{p\big(x^{(2)}|V\big)}{p\big(x^{(2)}\big)}\right)\left(\frac{p\big(x^{(2)}|V, C_2\big)}{p\big(x^{(2)}|V\big)}\right)\left(\frac{p\big(x^{(2)}|V\big)}{p\big(x^{(2)}|V, \mathcal{E}\big(x^{(1)}\big)\big)}\right). \tag{6}$$

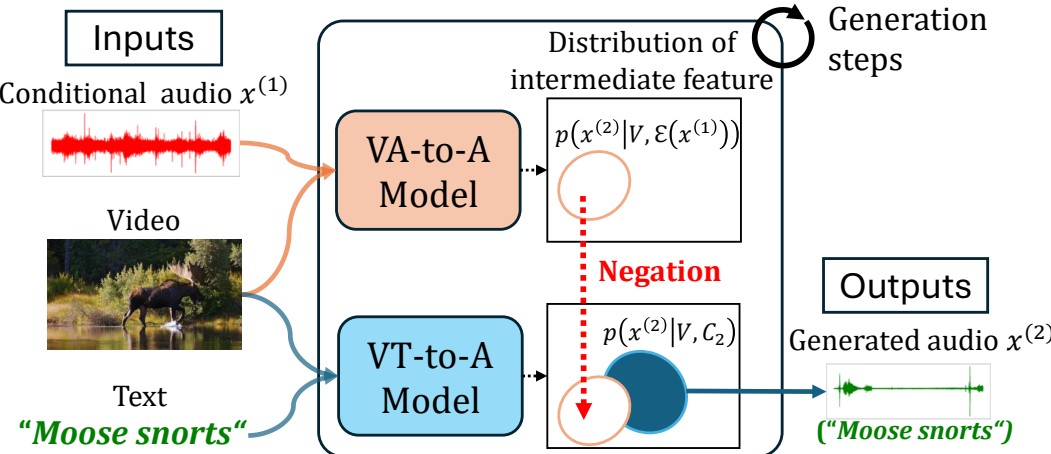

Figure 2: Overview of the proposed method. Each audio track should represent a distinct audio event. Using previously generated audio tracks as a condition for the negation concept, we explicitly push the current generation process away from the audio tracks already generated.

Similar to the derivation of Eq. (2), we can derive the guidance process based on this decomposition, as shown in the next section. It indicates we can sample $x^{(2)}$ using this new guidance with flow matching models. Iterating this process enables step-by-step generation in video-to-audio synthesis.

## 4 IMPLEMENTATION

### 4.1 GUIDED FLOW FOR STEP-BY-STEP GENERATION

The decomposition shown in Eq. (6) yields a new guidance formulation comprising four terms: one unconditional flow term and three guidance terms. However, adjusting the coefficients of the three guidance terms is cumbersome in practice. Given the empirical results (Kushwaha and Tian, 2025), where simplifying the guidance by removing $u_\theta(x_t, t, V, \varnothing)$ in Eq. (2) performs well, we also set the guidance coefficients so that $u_\theta(x_t, t, V, \varnothing)$ cancels out for simplification. Specifically, we put the sum of the coefficients of the first and third guidance terms to equal the coefficient of the second guidance term (see Section B for more details). This leads to the following guided flow:

$$\tilde{u}_{\theta,\psi}(x_t) = u_\theta(x_t, t, \varnothing, \varnothing) + \alpha(u_\theta(x_t, t, V, C_2) - u_\theta(x_t, t, \varnothing, \varnothing))$$
$$+ \beta\Big(u_\theta(x_t, t, V, C_2) - u_{\theta,\psi}\Big(x_t, t, V, \varnothing, x^{(1)}\Big)\Big), \tag{7}$$

where $\alpha$ and $\beta$ are the coefficients of the guidance terms. As we require an audio-conditioned flow in the last term, we introduce an additional set of trainable parameters $\psi$ to adapt the text-conditioned video-to-audio model for this prediction, as detailed in the next subsection.

The second term on the right-hand side of Eq. (7) corresponds to a standard guidance term in text-conditioned video-to-audio models, which appeared in Eq. (3). It strengthens the fidelity of the generated audio to the conditional video and text prompt. The third term is a new guidance term appearing in our proposed method, which pushes the generated audio away from the conditional audio $x^{(1)}$. This prevents the already generated audio events from being re-generated in the current generation step, enabling step-by-step generation without overlapping audio events. Since $x^{(1)}$ is used similarly for a negative prompt, we refer to this new guidance as Negative Audio Guidance (NAG).

### 4.2 TRAINING FLOW ESTIMATOR FOR NEGATIVE AUDIO GUIDANCE

All the flows appearing on the right-hand side of Eq. (7), except for the last one, can be estimated using standard text-conditional video-to-audio models. The remaining term is a conditional flow corresponding to the distribution $p(x^{(2)}|V, \mathcal{E}(x^{(1)}))$. As $\mathcal{E}(\cdot)$ is a standard type of audio conditioning, this flow estimator can be seen as an extended version of the video-to-audio model, enhanced by

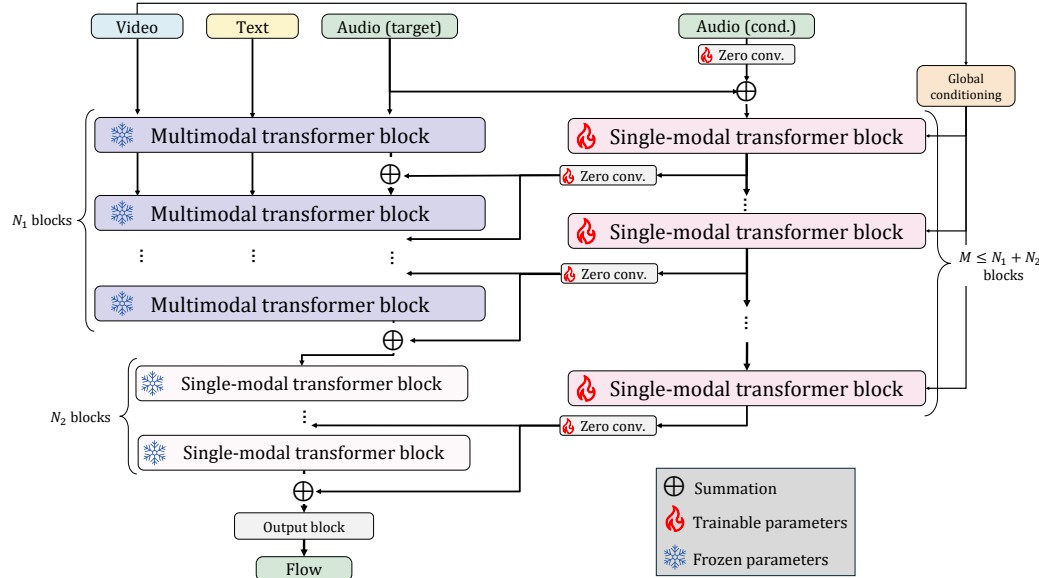

Figure 3: Overview of network architecture for the audio-conditional flow estimator. We adopt ControlNet for the multi-modal diffusion transformer (MM-DiT) to incorporate audio condition into the pre-trained MMAudio.

incorporating conditional audio as an additional input. Therefore, we train the flow estimator using ControlNet (Zhang et al., 2023) parameterized by $\psi$. As a base video-to-audio model, we used MMAudio, parameterized by $\theta$, for its high capability in video-to-audio synthesis.

**Model architecture.** Figure 3 shows an overview of the ControlNet architecture of the flow estimator. Since MMAudio uses a sophisticated architecture extended from MM-DiT (Esser et al., 2024), we adapt the architecture of ControlNet accordingly. Inspired by Stable Diffusion 3.5 (Stability-AI, 2024), we stack several single-modal transformer blocks to extract features from the conditional audio, and the features extracted at each block are added to the intermediate features of the corresponding blocks in MMAudio. During training, we freeze the pre-trained parameters of MMAudio and only update the parameters of the additional modules. See Section D for more details of the training.

**Training dataset.** We follow the training strategy of MMAudio, jointly using both text-video-audio and text-audio paired datasets for training. Specifically, we used VGGSound (Chen et al., 2020a) as a text-video-audio dataset, while Clotho (Drossos et al., 2020), AudioCaps (Kim et al., 2019), and WavCaps (Mei et al., 2024) were used as text-audio datasets. From each audio clip, we sampled a four-second audio segment as $x^{\text{tgt}}$ and another one as $x^{\text{cond}}$ so that the two segments do not overlap. We also extracted a video segment corresponding to $x^{\text{tgt}}$ as a conditional video $V$ when the data came from VGGSound; otherwise, we set the pretrained empty token of MMAudio as $V$. Then, the sampled clips were used to compute the flow $u_{\theta,\psi}\left(x_t^{\text{tgt}}, t, V, \varnothing, x^{\text{cond}}\right)$, and $\psi$ is optimized by minimizing the flow-matching loss.

## 5 EXPERIMENTS

### 5.1 EVALUATION SETUP

**Multi-Caps VGGSound: multi-captioned audio-video dataset for evaluation.** We constructed a new audiovisual dataset called Multi-Caps VGGSound to evaluate the step-by-step video-to-audio generation. We generated five captions using Qwen2.5-VL (Bai et al., 2025) for each video in the test split of the VGGSound dataset, comprising 15,221 video clips in total. We instructed the model to generate captions, each describing a different sound event that could appear in the input video's audio tracks, including both foreground and background audio. Since Qwen2.5-VL does not accept

audio as input, these captions were created solely based on the visual input without considering the original audio of the input video. Please refer to Section C in the appendix for more details.

**Task setup: Step-by-step audio generation.** We generated five audio tracks $\{x^{(k)}|k \in \{1, 2, \ldots 5\}\}$ corresponding to audio captions $\{C_k|k \in \{1, 2, \ldots 5\}\}$ for each video $V$ in the Multi-Caps VGGSound dataset. The generation order is determined based on the semantic similarity between the video and the caption, so that the model generates core events first (See Section G in the appendix for more details). We took the first eight-second segment from the video and generated sounds for this segment using different captions. Given the multiple generated audio tracks, we synthesized a composite audio $\tilde{x}$ by $\tilde{x} = \text{normalize}(\Sigma_k x^{(k)})$. We employed loudness normalization (Steinmetz and Reiss, 2021) as a simple mixing strategy for the composition, ensuring that the total loudness remained consistent with that of natural audio. The target loudness was set to -20 LUFS, which corresponds to the mean loudness of the VGGSound test set.

**Step-by-step audio generation with NAG.** To generate the audio tracks step-by-step using NAG, we used the composite audio of all audio tracks generated in the previous generation steps as a condition for NAG. Specifically, at the generation step for $x^{(k)}$, we synthesized a composite audio $\tilde{x}_{:k} = \text{normalize}(\Sigma_l^{k-1} x^{(l)})$ for the condition. We generated the first audio only using the standard classifier-free guidance in Eq. (3), as no audio track had been generated at the first generation step. For the guidance coefficients, we empirically set $\alpha = 4.5$ and $\beta = 1.5$ in Eq. (7) (see Section E for more details).

**Baseline models.** We compared our proposed method with several open-sourced text-and-video conditional audio (TV2A) generation models. We chose each State-of-the-Art TV2A model among various training approaches: Seeing-and-Hearing (Xing et al., 2024) as a TV2A model adapted from the T2A model in a zero-shot manner, FoleyCrafter (Zhang et al., 2024) as a TV2A model adapted from the T2A model through fine-tuning, and MMAudio (Cheng et al., 2025) as a TV2A model trained from scratch. Note that our model is built upon MMAudio with additional audio conditions introduced by the proposed ControlNet architecture. We compared the proposed NAG to the original MMAudio-S-16k model with the classifier-free guidance (CFG) or negative prompting.

We tested two generation processes to generate multiple audio tracks and obtain a composite audio using the baseline models: independent generation and step-by-step generation based on negative prompts. In the independent generation, we generated five sounds for each video using different text conditions. In this case, each generation process does not access the other audio tracks or their corresponding captions. In the step-by-step generation based on negative prompts, we generated each audio track with negative prompting (Woolf, 2022) to ensure it was distinct from all the captions used in the previous generation steps. Specifically, for the video $V$ with the $k$-th audio caption $C_k$, we computed the guided flow by $\tilde{u}_\theta(x_t) = u_\theta(x_t, t, \varnothing, \varnothing) + w_1(u_\theta(x_t, t, V, C_k) - u_\theta(x_t, t, \varnothing, C_{k,\text{neg}}))$ at each timestep, where $C_{k,\text{neg}}$ is the concatenation of the other captions $\{C_l|l < k\}$.

**Evaluation metrics.** We assessed the quality of both the composite audio and the individual audio tracks to evaluate the step-by-step audio generation.

Following the prior work (Cheng et al., 2025), we evaluated the composite audio in terms of audio quality, semantic alignment, and temporal alignment. We assessed the audio quality of the generated audio using Fréchet Distance (FD), Kullback–Leibler (KL) distance, and Inception Score (IS) (Salimans et al., 2016). We used PANNs (Kong et al., 2020) ($\text{FD}_{\text{PANNs}}$) and VGGish (Gemmeke et al., 2017) ($\text{FD}_{\text{VGG}}$) for computing FD, and PANNs ($\text{KL}_{\text{PANNs}}$) and PaSST ($\text{KL}_{\text{PaSST}}$) for computing KL, and PANNs for computing IS, respectively. We assessed the semantic alignment between the input video and the composite audio by the cosine similarity between their embeddings extracted by ImageBind (Girdhar et al., 2023) (IB-score). We assessed the temporal alignment between the input video and the generated audio with Synchformer (Iashin et al., 2024) (DeSync), where we took two 4.8-second segments at the beginning and end and averaged their scores.

For the evaluation of each audio track, we assessed its quality from four aspects: audio separability between the audio tracks generated for the same video, audio quality, audio-text alignment, and audio-video alignment of each audio track. Since distinct audio components should be represented in separate audio tracks, each generated audio track should differ from the other tracks. To evaluate

| Method | Audio Quality | | | | | Semantic Align. | Temporal Align. |
|---|---|---|---|---|---|---|---|
| | $FD_{PANNs}\downarrow$ | $FD_{VGG}\downarrow$ | $KL_{PANNs}\downarrow$ | $KL_{PaSST}\downarrow$ | IS$\uparrow$ | IB-score$\uparrow$ | DeSync$\downarrow$ |
| **One-Step Generation with Fused Caption (Reference as no separated audio track is available)** | | | | | | | |
| Seeing-and-Hearing | 25.42 | 5.68 | 2.81 | 2.76 | 6.45 | 36.88 | 1.22 |
| FoleyCrafter | 16.93 | 2.29 | 2.60 | 2.52 | 11.93 | 27.78 | 1.23 |
| MMAudio-S-16k | 6.75 | 1.03 | 2.09 | 2.04 | 13.66 | 29.45 | 0.46 |
| **Independent Generation** | | | | | | | |
| Seeing-and-Hearing | 31.81 | 7.68 | 3.10 | 2.65 | 4.12 | 20.16 | 1.19 |
| FoleyCrafter | 20.04 | 3.23 | 2.70 | 2.36 | 9.21 | 25.26 | 1.18 |
| MMAudio-S-16k | 7.76 | 1.35 | 2.02 | 1.84 | 10.42 | 28.13 | **0.42** |
| **Step-by-Step Generation with Negative Prompting** | | | | | | | |
| FoleyCrafter | 22.34 | 4.64 | 2.94 | 2.47 | 6.02 | 18.83 | 1.19 |
| MMAudio-S-16k | 9.21 | 1.77 | 2.15 | 1.89 | 9.08 | 25.89 | 0.45 |
| **Step-by-Step Generation with Negative Audio Guidance** | | | | | | | |
| Ours | **6.47** | **0.98** | **2.01** | **1.76** | 10.58 | **28.65** | **0.42** |

Table 1: Quantitative evaluation of the composite audio synthesized from the generated multiple audio tracks. The results of one-step generation using a fused caption are shown as a reference.

| Method | Audio Separability | Audio Quality | A-T Align. | A-V Align. |
|---|---|---|---|---|
| | CLAP A-A$\downarrow$ | IS$\uparrow$ | CLAP T-A$\uparrow$ | IB-score$\uparrow$ |
| MMAudio-S-16k | 79.75 | **12.47** | 28.36 | **27.76** |
| MMAudio-S-16K + Neg. Prompting | 75.57 | 11.19 | 27.14 | 24.53 |
| MMAudio-S-16K + NAG (Ours) | **71.38** | 12.01 | **28.91** | 26.67 |

Table 2: Quantitative evaluation of individual audio tracks. Our proposed method successfully improves audio separability among multiple tracks while maintaining other scores.

audio separability, we computed the similarity between the CLAP (Wu et al., 2023) audio embeddings for each pair of audio tracks (10 pairs per video). For Audio-Text alignment, we computed the similarity between CLAP text embeddings from the input prompts (used to generate each audio track) and the corresponding CLAP audio embeddings. For audio quality and audio-video alignment, we adopted IS and IB-score, respectively, as in the composite audio evaluation protocol.

## 5.2 MAIN RESULTS

**Objective evaluation on the composite audio.** Table 1 shows the quantitative evaluation of the composite audio. Our proposed method achieves the best results for all metrics except IS among all the methods. We also evaluated the baseline models' one-step generation with the caption created by fusing the five captions for each video. Though this generation process does not provide each audio track and differs from our goal of step-by-step generation, these values indicate the best possible scores of each baseline model.

**Objective evaluation on each audio track.** Table 2 shows the quantitative evaluation of the individual audio tracks. The vanilla MMAudio-S-16k with CFG struggles to generate well-separated sounds for each audio track, which is reflected in a lower audio separability score, although it achieves high audio quality and A-V Alignment. Using negative prompting improves the audio separability but drastically degrades all the other scores. Using NAG also successfully improves the audio separability while maintaining high audio quality and A-V alignment. The A-T alignment score marginally improves from that of the vanilla MMAudio. We hypothesize that less contamination of other audio concepts results in a better A-T alignment score.

**Visual comparison with baselines.** Figure 4 visually compares these three methods. The first audio track is the same in all methods, since it is generated by using only CFG. The vanilla MMAudio-S-16k tends to generate similar audio for multiple audio tracks. All generated audio tracks by the vanilla

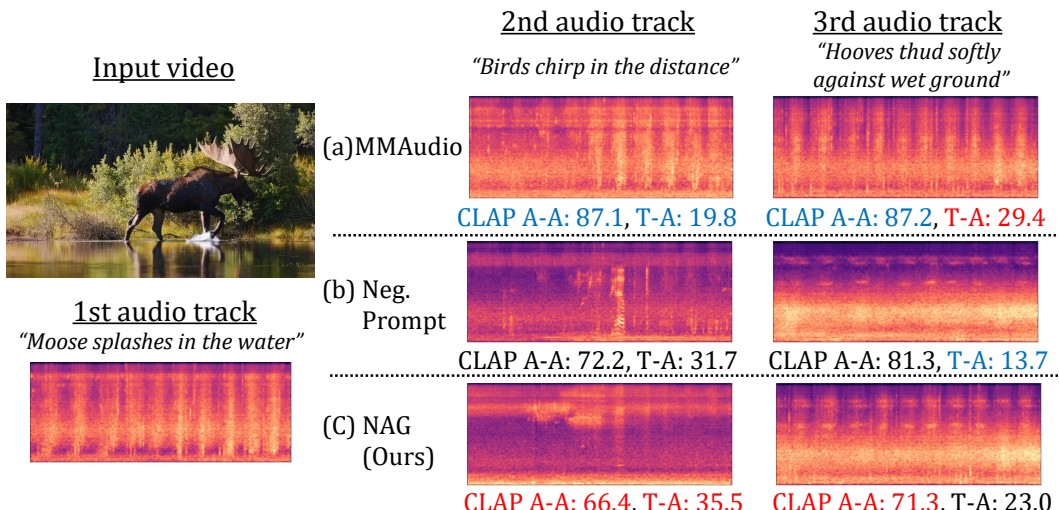

Figure 4: Spectrogram visualizations of step-by-step audio generation using (a) vanilla MMAudio, (b) MMAudio with negative prompting, and (c) MMAudio with Negative Audio Guidance (NAG). The best and worst CLAP A-A and T-A scores are highlighted in red and blue, respectively. The first audio track is generated using (a) in all settings, resulting in identical outputs. Our proposed method effectively suppresses previously generated sounds in subsequent steps (second and third tracks) while maintaining high alignment with the target text prompts.

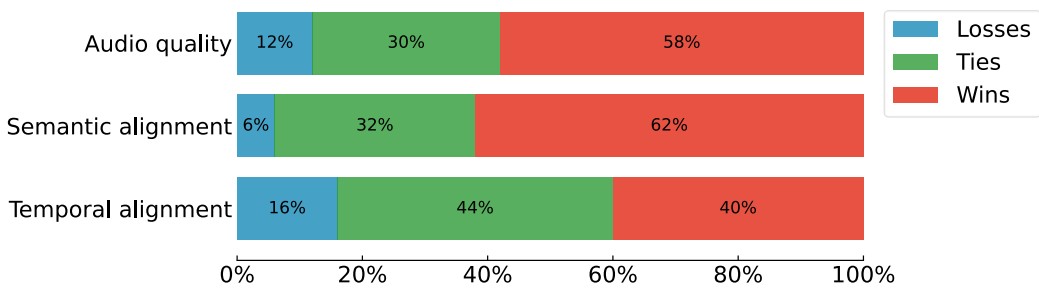

Figure 5: Results of user preference comparison between baseline (MMAudio-S-16k) and our method (MMAudio-S-16k with NAG) for composite audio. "Wins" indicates the percentage of users who choose the composite audio generated by our method.

MMAudio-S-16k contain the sound of water as the moose walks (visually shown as vertical segments that appear at regular intervals). Since the moose and its movement are prominent in the input video, MMAudio tends to include the sound related to them regardless of the input text prompt. This is also reflected in the higher IB-score, indicating that all audio tracks are semantically aligned with the input video, regardless of whether the input prompt represents background audio (as in the case of the second audio track). Using negative prompting suppressed this contamination of the audio content, but it tends to suffer from worse text alignment. In contrast, the proposed NAG successfully suppressed the audio components already generated in the early generation steps while achieving better text alignment. It generates a missing sound specified by the text prompt by explicitly making the generation process away from the already generated sounds. Please refer to Section K in the appendix for more generated samples.

**Subjective evaluation.** We also conducted a user study for subjective evaluation. Figure 5 shows the results of the user study for the final composite audio, demonstrating that our method is preferred in terms of audio quality, semantic alignment, and temporal alignment compared to the baseline. Please refer to Section A in the appendix for the results of each generated audio track, more detailed statistical analysis, and the details of this user study setup.

## 6 CONCLUSION

We introduced a novel video-to-audio generation method, guided by text, video, and audio conditions, to enable step-by-step synthesis. By applying negative audio guidance alongside a text prompt, our approach generates multiple well-separated audio tracks for the same video input, facilitating high-quality composite audio synthesis. Importantly, our method does not require specialized training datasets. We built it on a pre-trained video-to-audio model by adapting ControlNet for audio conditioning, which can be trained on accessible datasets. Quantitative and subjective evaluation shows that our method improves separability and text fidelity of generated audio at each step, and improves the quality of the final composite audio.

## ETHICS STATEMENT

This work uses only publicly available audiovisual datasets for training. It does not contain sensitive or personally identifiable information, and no additional human subjects were involved. Our approach avoids reliance on costly multi-reference datasets, promoting accessibility and reproducibility.

While the method is intended for academic and creative research, we acknowledge potential misuse of generative audio technologies. To reduce such risks, careful dataset selection and robust filtering or moderation should be applied in downstream use.

## REPRODUCIBILITY STATEMENT

The proofs of the theoretical formulations of our proposed method (Equations (6) and (7)) are shown in Section B in the appendix. Section 4 describes the implementation details of guided flow computation, including the inference and training processes, model architecture (visualized in Figure 3), and the training dataset. The dataset construction pipeline and the training setup (including hyperparameters and computational resources) are further elaborated in Sections C and D, respectively. Evaluation setup and metrics are detailed in Section 5.1. We will open-source our code and dataset upon acceptance.

## THE USE OF LARGE LANGUAGE MODELS (LLMS)

We utilized (V)LLMs for dataset construction to add multiple plausible captions for videos (please refer to the Section C in the appendix for details). We also used LLMs for academic proofreading. However, all research ideas were developed solely by the authors.

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

CONTENTS

# A USER STUDY

We conducted a user study to perform a subjective assessment on the Multi-Caps VGGSound dataset. We used independent generation with MMAudio-S-16k as the baseline, corresponding to the case without negative audio guidance ($\beta = 0$ in Eq. (9)), to assess the effectiveness of our proposed method. We randomly sampled five video-caption sets from the dataset (each video has five captions) and generated five audio tracks for each video using the baseline and the proposed method. As described in Section 5, we then synthesized composite audio for each video by mixing generated tracks, followed by loudness normalization. In total, we showed 60 videos to each evaluator (50 videos with individual audio tracks and 10 videos with composite audio). Human evaluators were asked to assess the quality of both individual audio tracks and the composite audio.

For the evaluation of individual audio tracks, evaluators rated each track on a scale from 1 to 5 (1-5; Poor, Subpar, Fair, Good, Excellent) across the following three aspects:

1. **Separability**: High if the audio does not contain content already present in previous audio tracks.

2. **Audio quality**: High if the audio is free from noise, distortion, or artifacts.

3. **Text fidelity**: High if the audio accurately reflects the caption.

For the evaluation of composite audio, we performed A/B testing on pairs of composite audios, one generated by the baseline and the other by our method. Specifically, five pairs of composite audios (each corresponding to the same video) were presented to evaluators, who were asked the following three questions for each pair:

1. **Audio quality**: Which audio is of higher quality?

2. **Semantic alignment**: Which audio has better semantic alignment with the video?

3. **Temporal alignment**: Which audio has better temporal alignment with the video?

For each question, evaluators could choose from three response options: "Audio A is better", "Audio B is better", and "Neutral".

We collected 400 responses for the individual audio tracks (we omitted an evaluation for the first track since the first track is identical between the two methods) and 50 responses for the composite audio from 10 evaluators. To verify statistical significance, we compute 95% confidence intervals (CI) using the Wilson score interval for preference data and the standard error for rating scores. For the preference CI, ties are evenly split between wins and losses. Table A2 shows the results for the individual audio tracks. Our method received significantly higher ratings for separability, and marginally higher ratings for both audio quality and text fidelity. Figure 5 and table A1 show the results for the composite audio. Overall, the proposed method was preferred equally or more across all evaluation criteria.

|  | Audio quality↑ | Semantic alignment↑ | Temporal alignment ↑ |
|---|---|---|---|
| Win rate | **71.36** $\pm 7.71$ | **76.00** $\pm 7.62$ | **61.14** $\pm 7.85$ |

Table A1: The win rates of our proposed method (MMAudio-S-16k + NAG) against MMAudio-S-16k for composite audio. 95% confidence intervals are also reported as $\pm X$.

| Method | Separability↑ | Audio quality↑ | Text fidelity ↑ |
|---|---|---|---|
| MMAudio-S-16k | 2.24$\pm 0.15$ | 2.89$\pm 0.14$ | 2.42$\pm 0.18$ |
| MMAudio-S-16k + NAG (Ours) | **3.35**$\pm 0.15$ | **3.30**$\pm 0.14$ | **3.12**$\pm 0.18$ |

Table A2: Average ratings for individual audio tracks generated by the baseline and our method. Mean and 95% confidence interval are reported for each aspect.

## B  DETAILS OF THE FORMULATION

PROOF OF THE EQUATION (6)

Recall that the target distribution of each generation step can be written as:

$$p\left(x^{(2)}\Big|V,C_2,\bar{\mathcal{E}}\left(x^{(1)}\right)\right) \propto p\left(x^{(2)},V,C_2,\bar{\mathcal{E}}\left(x^{(1)}\right)\right)$$

$$\propto p\left(x^{(2)},V\right)p\left(C_2\Big|x^{(2)},V\right)p\left(\mathcal{E}\left(x^{(1)}\right)\Big|x^{(2)},V\right)^{-1}$$

$$= p\left(x^{(2)}\right)p\left(V\Big|x^{(2)}\right)p\left(C_2\Big|x^{(2)},V\right)p\left(\mathcal{E}\left(x^{(1)}\right)\Big|x^{(2)},V\right)^{-1}. \quad \text{(A1)}$$

Using Bayes's theorem, we can decompose the last three terms in Eq. (A1) as follows:

$$p\left(V\Big|x^{(2)}\right) = \frac{p(x^{(2)}|V)p(V)}{p(x^{(2)})}$$

$$\propto \frac{p(x^{(2)}|V)}{p(x^{(2)})}, \quad \text{(A2)}$$

$$p\left(C_2\Big|x^{(2)},V\right) = \frac{p(x^{(2)}|V,C_2)p(C_2|V)}{p(x^{(2)}|V)}$$

$$\propto \frac{p(x^{(2)}|V,C_2)}{p(x^{(2)}|V)}, \quad \text{(A3)}$$

$$p\left(\mathcal{E}\left(x^{(1)}\right)\Big|x^{(2)},V\right) = \frac{p(x^{(2)}|\mathcal{E}(x^{(1)}),V)p(\mathcal{E}(x^{(1)})|V)}{p(x^{(2)}|V)}$$

$$\propto \frac{p(x^{(2)}|V,\mathcal{E}(x^{(1)}))}{p(x^{(2)}|V)}. \quad \text{(A4)}$$

Note that we omit terms unrelated to $x^{(2)}$, since $x^{(2)}$ is the generation target. Substituting Eqs. (A2), (A3), and (A4) into Eq. (A1), we get:

$$p\left(x^{(2)}\Big|V,C_2,\bar{\mathcal{E}}\left(x^{(1)}\right)\right)$$

$$\propto p\left(x^{(2)}\right)\left(\frac{p(x^{(2)}|V)}{p(x^{(2)})}\right)\left(\frac{p(x^{(2)}|V,C_2)}{p(x^{(2)}|V)}\right)\left(\frac{p(x^{(2)}|V)}{p(x^{(2)}|V,\mathcal{E}(x^{(1)}))}\right). \quad \text{(A5)}$$

Therefore, Eq. (6) holds.

DERIVATION OF THE NEGATIVE AUDIO GUIDANCE IN EQUATION (7)

Similar to the guided flow proposed by Kushwaha and Tian (2025), we can derive the guided flow corresponding to Eq. (6) (or identically Eq. (A5)) as follows:

$$\tilde{u}_{\theta,\psi}(x_t) = u_\theta(x_t,t,\varnothing,\varnothing) + w_1'(u_\theta(x_t,t,V,\varnothing) - u_\theta(x_t,t,\varnothing,\varnothing))$$

$$+ w_2'(u_\theta(x_t,t,V,C_2) - u_\theta(x_t,t,V,\varnothing))$$

$$+ w_3'\Big(u_\theta(x_t,t,V,\varnothing) - u_{\theta,\psi}\left(x_t,t,V,\varnothing,x^{(1)}\right)\Big), \quad \text{(A6)}$$

where $w_1'$, $w_2'$, and $w_3'$ are the coefficients of the guidance terms. The four terms on the right-hand side of Eq. (A6) respectively correspond to the four factors on the right-hand side of Eq. (A5).

Following the empirical results provided by Kushwaha and Tian (2025), we consider canceling out $u_\theta(x_t, t, V, \varnothing)$ for simplification. Specifically, we set $w_1' = \alpha$, $w_3' = \beta$, and $w_2' = \alpha + \beta$ as follows:

$$\tilde{u}_{\theta,\psi}(x_t) = u_\theta(x_t, t, \varnothing, \varnothing) + \alpha\Big(\cancel{u_\theta(x_t, t, V, \varnothing)} - u_\theta(x_t, t, \varnothing, \varnothing)\Big)$$
$$+ (\alpha + \beta)\Big(u_\theta(x_t, t, V, C_2) - \cancel{u_\theta(x_t, t, V, \varnothing)}\Big)$$
$$+ \beta\Big(\cancel{u_\theta(x_t, t, V, \varnothing)} - u_{\theta,\psi}\Big(x_t, t, V, \varnothing, x^{(1)}\Big)\Big), \qquad \text{(A7)}$$

which yields Eq. (7).

## C  DETAILS OF THE MULTI-CAPS VGGSOUND DATASET

As described in Section 5.1, we generated five captions for each video in the test split of the VGGSound dataset using Qwen2.5-VL (Bai et al., 2025), resulting in 76,105 video-caption pairs (15,221 videos $\times$ 5 captions). Figure A1 shows an overview of the dataset construction workflow.

We adopted a two-step approach to ensure that captions follow a unified format across all videos (i.e., short, simple sentences that describe distinct audio events). Specifically, given an input video, we first generated multiple possible free-form audio captions describing the audio events likely present in the video. Next, we reformatted the output into a structured JSON format using the generation of structured outputs (as implemented in vLLM (Kwon et al., 2023)). The full prompt we used in the first step is shown in Fig. A2. While Qwen2.5-VL generates multiple captions in response to this prompt, the output may vary between inferences. To standardize this, the second step converts the results into a unified JSON format that lists only the captions. This structured format is well-suited for use as text conditions in text-conditional video-to-audio models. Examples of video and caption pairs are shown in Figure A3.

Using VLM-generated captions may introduce some noise, as not all descriptions perfectly match the actual audio. However, we manually inspected a subset and confirmed that the captions remain visually plausible. Since Foley workflows typically begin with a silent video and incrementally add plausible sounds based solely on visual context, this setup naturally aligns with Foley-style use cases in which the original recording does not match the target audio.

## D  DETAILS OF MODEL ARCHITECTURE, TRAINING, AND INFERENCE

We added one transformer block in the ControlNet for every two blocks in the main network (i.e., $N_1 + N_2 = 12$ and $M = 6$ in Fig. 3). We set the channel dimensions and number of heads for multi-head attention to match the settings of MMAudio-S-16k. Our ControlNet has a total of 107M parameters, and the generation at each step, computed by this ControlNet using NAG, takes 2.07 seconds on an H100.

We followed the training setup of MMAudio (Cheng et al., 2025) for training the ControlNet. We used the AdamW optimizer with a learning rate of $10^{-4}$, $\beta_1 = 0.9$, $\beta_2 = 0.95$, and a weight decay of $10^{-6}$. The network is trained for 200K iterations with a batch size of 512. Compared to MMAudio's default of 300K iterations, we reduced the number of training steps to 200K, as we observed earlier convergence in our experiments. We only updated the parameters of the ControlNet while fixing the pre-trained parameters of MMAudio, enabling more efficient training. For learning rate scheduling, we applied a linear warm-up over the first 1K steps up to $10^{-4}$, followed by two reductions, each by a factor of 10, after 80% and 90% of the total training steps. We used mixed-precision training with `bf16` for the training efficiency and trained on 8 H100 GPUs. The entire training process, including evaluation on the validation and test sets every 20K iterations, took approximately 10 hours. After training, we applied post-hoc EMA (Karras et al., 2024) with a relative width $\sigma_{\text{rel}} = 0.05$ to obtain the final parameters of the ControlNet.

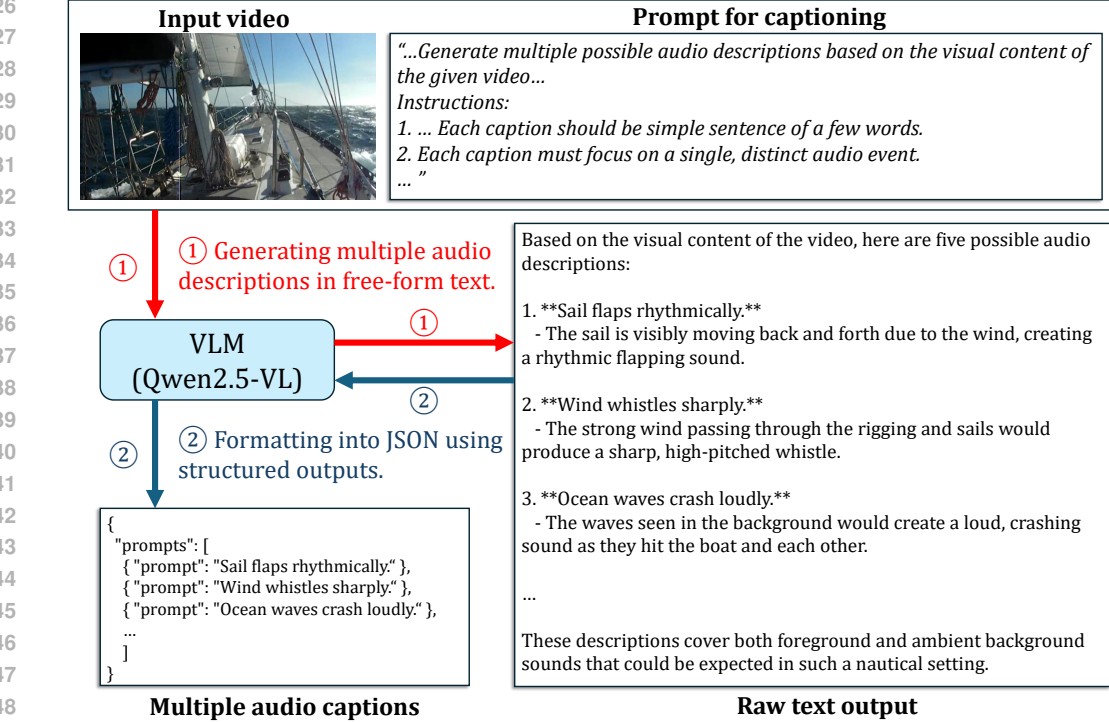

Figure A1: Overview of the dataset construction pipeline. Multiple audio captions were generated for each video using Qwen2.5-VL via a two-step process: free-form captioning followed by structured JSON formatting. The input prompt on the top is simplified; see Fig. A2 for the full version.

> " *Task:*
> *You are a professional sound effects creator. Generate multiple possible audio descriptions based on the visual content of the given video. Each description should focus on a single, distinct audio event, and each could be either a foreground sound or an ambient background sound. Foreground sounds are the sounds that are directly depicted in the video (e.g., a dog barking, footsteps). Ambient background sounds are the sounds that could be inferred or imagined from the video's context but not explicitly shown (e.g., distant wind, soft city hum).*
>
> *Examples:*
> *#1 The dog barks loudly.*
> *#2 The river flows gently.*
>
> *Instructions:*
> *1. Use the format: 'Noun + Verb + Adverbs' (adverbs are optional). Each caption should be simple sentence of a few words.*
> *2. Each caption must focus on a single, distinct audio event.*
> *3. Begin each caption with '#N', where N is the index of the description.*
> *4. AVOID DUPLICATES, and provide up to 5 descriptions per video.*"

Figure A2: Full prompt for generating multiple possible audio captions.

# E    SENSITIVITY ANALYSIS OF THE NEGATIVE AUDIO GUIDANCE COEFFICIENTS

We conducted a sensitivity study on the guidance coefficients of NAG ($\alpha$, $\beta$ in Eq. 7). Specifically, we varied $\alpha \in \{3.5, 4.5\}$ and $\beta \in \{0.0, 1.0, 1.5, 2.0\}$ and generated audio tracks for all combinations of these parameter pairs in **random generation order**. The individual audio tracks and their corresponding composite audio were evaluated using the same setup and metrics described in Section 5.1. We used CLAP A-A for audio separability, IS and $FD_{PANNs}$ for audio quality, CLAP T-A for text fidelity, IB-score for semantic alignment with video, and DeSync for temporal alignment with video.

| **Video** | **Generated captions** |
|---|---|
| 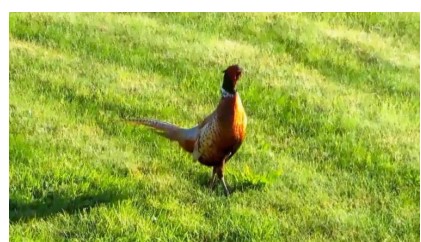 | 0: *"Ocean waves crash loudly."*
1: *"Sail flaps rhythmically."*
2: *"Wind whistles sharply."*
3: *"Ropes creak softly."*
4: *"Metal clanks intermittently."* |
| 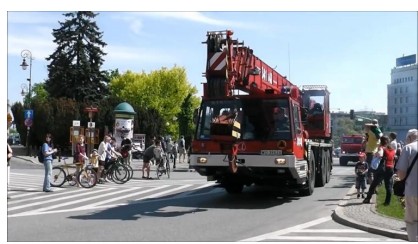 | 0: *"Pheasant cackles softly."*
1: *"Grass rustles gently underfoot."*
2: *"Wings flutter faintly."*
3: *"Birds chirp in the distance."*
4: *"Wind whispers through the grass."* |
| | 0: *"Engine roars powerfully."*
1: *"Wheels grind heavily on pavement."*
2: *"Horn blasts loudly."*
3: *"People chatter excitedly."*
4: *"Bicycles click softly as they move."* |

Figure A3: Examples of the Multi-Caps VGGSound dataset. We added multiple captions to the test split of the VGGSound dataset using Qwen2.5-VL, shown in Figure A1.

The results are summarized in Table A3. Both CLAP A-A and CLAP T-A for the individual audio tracks consistently improved with increasing $\beta$. This indicates that NAG effectively generates well-separated audio tracks with enhanced text alignment, likely due to reduced contamination from other audio concepts. The best $FD_{PANNs}$ and IB-score are achieved at $\beta = 1.0$, indicating that a moderate strength of NAG also enhances audio fidelity and semantic alignment with video. Using a small $\alpha$ slightly deteriorates performance across all metrics, potentially due to the degradation of the performance of the base MMAudio (the default CFG strength recommended by Cheng et al. (2025) is 4.5). Considering trade-offs among these metrics, we selected $\alpha = 4.5$ and $\beta = 1.5$ as our default setting.

## F ONE-STEP GENERATION WITH POST-PROCESSING

To obtain multiple separated audio tracks, one possible approach is a one-step generation with post-processing: generating a composite audio from a fused caption, followed by track decomposition using audio source separation. However, this approach fails for two reasons: (i) **one-step generation frequently misses audio events** when the text prompt contains multiple concepts, and (ii) **audio source separation degrades quality**, producing artifacts and unstable separated outputs. We compare this approach with our proposed method to clarify its limitations.

We first generate single-track outputs using MMAudio-S-16k with fused captions (the composite sound results are shown in Table 1). Then, we apply AudioSep (Liu et al., 2022b; 2023) to obtain separated audio tracks. We use the official AudioSep model with its default configuration[2] and the same captions for separation as in our step-by-step approach.

---

[2]https://github.com/Audio-AGI/AudioSep

| Method | Individual Audio Tracks | | | Composite Audio | | | |
|---|---|---|---|---|---|---|---|
| | CLAP A-A↓ | IS↑ | CLAP T-A↑ | FD$_{PANNs}$↓ | IS↑ | IB-score↑ | DeSync↓ |
| $\alpha = 3.5, \beta = 0.0$ | 79.46 | 12.27 | 28.34 | 7.83 | 10.37 | 27.80 | 0.45 |
| $\alpha = 3.5, \beta = 1.0$ | 75.22 | 12.06 | 28.75 | 7.52 | 10.17 | 27.94 | 0.45 |
| $\alpha = 3.5, \beta = 1.5$ | 73.45 | 11.84 | 28.89 | 7.62 | 9.90 | 27.62 | 0.45 |
| $\alpha = 3.5, \beta = 2.0$ | 71.97 | 11.62 | 29.01 | 7.84 | 9.69 | 27.28 | 0.45 |
| $\alpha = 4.5, \beta = 0.0^*$ | 79.75 | 12.47 | 28.36 | 7.76 | 10.42 | 28.13 | 0.43 |
| $\alpha = 4.5, \beta = 1.0$ | 76.21 | 12.22 | 28.69 | 7.30 | 10.38 | 28.38 | 0.43 |
| $\alpha = 4.5, \beta = 1.5^\dagger$ | 74.74 | 12.00 | 28.79 | 7.32 | 10.21 | 28.15 | 0.43 |
| $\alpha = 4.5, \beta = 2.0$ | 73.44 | 11.80 | 28.88 | 7.52 | 9.92 | 27.88 | 0.43 |

Table A3: Sensitivity study on the guidance coefficients of NAG ($\alpha, \beta$ in Eq. (9)). We determined generation order by the random order strategy (see Section G and Table A5) and the same order is applied across all settings. The first three metrics are computed on the individual audio tracks, and the last four are computed on the composite audio. $*$: identical to the independent generation of MMAudio-S-16k with the default CFG strength of 4.5. $\dagger$: our default setting.

| Method | Audio Separability | Audio Quality | A-T Align. | A-V Align. |
|---|---|---|---|---|
| | CLAP A-A↓ | IS↑ | CLAP T-A↑ | IB-score↑ |
| MMAudio-S-16k (one-step) $\rightarrow$ AudioSep | 79.11 | 9.04 | 23.42 | 21.96 |
| MMAudio-S-16k + NAG (Ours, step-by-step) | **71.38** | **12.01** | **28.91** | **26.67** |

Table A4: Comparison between one-step generation with audio source separation and our method.

Table A4 shows a comparison between this approach and our method. The separated tracks exhibit substantially worse text-audio and video-audio alignment (CLAP T-A and IB-Score) because the one-step results often fail to capture all required sound events. These missing events lead to silent or noisy separated tracks, hurting both audio separability (CLAP A-A) and overall quality. These results indicate that one-step generation with post-processing is a suboptimal strategy for producing multiple audio tracks, whereas our sequential method effectively mitigates these issues. It is also worth noting that this approach complicates user interaction: revising only a specific sound event requires re-running both the generator and a separation model, which is unintuitive and fragile. In contrast, our method provides an intuitive user interface that lets a user focus on a specific audio event at a time.

## G  COMPARISON OF GENERATION ORDER

To study the effect of generation order, we ranked captions based on text-video similarity using the ImageBind score. We tested three variants: random order, descending order (where the core event is first), and ascending order (where the subtle event is first). The results are shown in Table A5. Descending order provides the best results for all metrics, indicating that generating the prominent event in the video first is vital to improve the generation quality in step-by-step generation.

## H  ANALYSIS ON NUMBER OF CAPTIONS

We used a fixed number of captions in the main experiments to ensure consistent evaluation. In real use cases, however, the number of sequential generation steps is chosen by the user based on the visual content and the desired number of audio events. To examine whether our method remains effective across a broader range of plausible audio events, we conducted additional experiments on the VGGSounder (Zverev et al., 2025) dataset. VGGSounder provides a flexible number of human-annotated captions per video, reflecting both the visual content and the ground-truth audio. We selected videos with 2–5 captions and evaluated the baseline model (MMAudio-S-16k) and our method (MMAudio-S-16k + NAG) on each subset separately.

| Generation order | Audio Quality | | | | | Semantic Align. | Temporal Align. |
|---|---|---|---|---|---|---|---|
| | $FD_{PANNs}\downarrow$ | $FD_{VGG}\downarrow$ | $KL_{PANNs}\downarrow$ | $KL_{PaSST}\downarrow$ | IS↑ | IB-score↑ | DeSync↓ |
| MMAudio-S-16k | 7.76 | 1.35 | 2.02 | 1.84 | 10.42 | 28.13 | **0.42** |
| Random | 7.32 | 1.24 | 2.02 | 1.78 | 10.21 | 28.15 | 0.43 |
| Ascending | 7.36 | 1.26 | 2.02 | 1.78 | 10.02 | 28.10 | 0.43 |
| Descending | **6.47** | **0.98** | **2.01** | **1.76** | **10.58** | **28.65** | 0.42 |

Table A5: Comparison of generation order. We rank the captions based on the text-video ImageBind score and sort them in ascending and descending order.

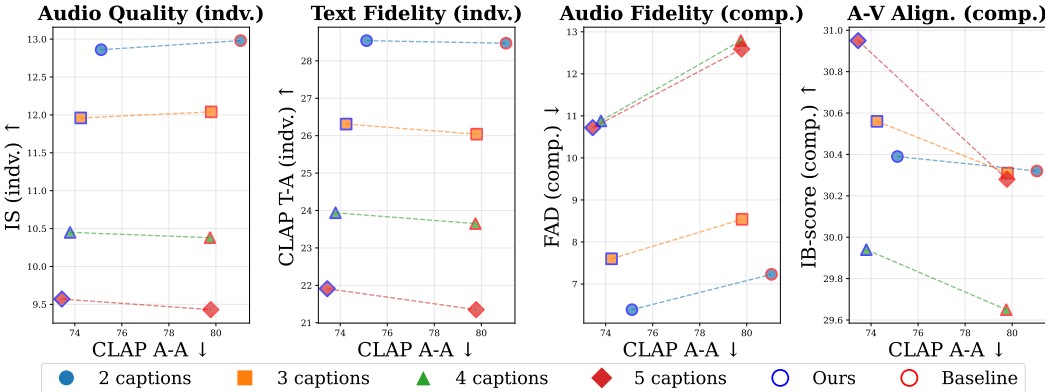

Figure A4: Analysis of the number of captions using VGGSounder. We compare MMAudio-S-16k (Baseline) and MMAudio-S-16k + NAG (Ours) on individual-track quality and text fidelity (left), and on composite-audio fidelity and audiovisual alignment (right). Audio separability is shown on the x-axis. Our method consistently improves separability without degrading quality, with larger gains as the number of captions increases.

Figure A4 summarizes the results. The two plots on the left report audio quality and text fidelity for the individual tracks, while the two plots on the right report audio fidelity and audiovisual alignment for the composite audio. All plots use the audio separability of each track on the x-axis. Across all caption counts, the baseline model exhibits consistently low separability, whereas our method reliably improves separability without sacrificing audio quality or text alignment. The improvement becomes more pronounced as the number of captions increases. For composite audio evaluation, our method consistently enhances both audio fidelity and audiovisual alignment. Overall, these results demonstrate that the proposed step-by-step generation is beneficial even when only a few sound events are present, and its advantage becomes increasingly significant as the number of audio events grows.

## I EVALUATION ON AUDIOCAPS AND MOVIE GEN AUDIO BENCH DATASETS

To validate cross-dataset generalizability, we additionally evaluate our method on AudioCaps (Kim et al., 2019) and Movie Gen Audio Bench (Polyak et al., 2024b). We follow the same caption-generation procedure as Multi-Caps VGGSound using Qwen2.5-VL to generate multiple captions (see Section C). The results are shown in Tables A6 and A7. Our method consistently improves audio separability and overall performance, demonstrating its generalizability across datasets.

## J LIMITATION

**Slight degradation of the audio quality of each audio track.** In terms of individual audio track quality, our method marginally improves text fidelity but slightly degrades audio quality. While NAG effectively eliminates contamination from other audio tracks, the outputs sometimes exhibit

| Method | Individual Audio Tracks | | | | Composite Audio | | |
|---|---|---|---|---|---|---|---|
| | CLAP A-A↓ | IS↑ | CLAP T-A↑ | IB-Score↑ | IS↑ | IB-score↑ | DeSync↓ |
| MMAudio-S-16k | 68.89 | **9.65** | 29.09 | **24.91** | 6.81 | 25.73 | 0.56 |
| MMAudio-S-16k + NAG (Ours) | **60.43** | 9.29 | **29.25** | 23.46 | **7.08** | **26.36** | **0.54** |

Table A6: Quantitative evaluation of both the composite audio and individual audio tracks generated from step-by-step generation on the AudioCaps test set.

| Method | Individual Audio Tracks | | | | Composite Audio | | |
|---|---|---|---|---|---|---|---|
| | CLAP A-A↓ | IS↑ | CLAP T-A↑ | IB-Score↑ | IS↑ | IB-score↑ | DeSync↓ |
| MMAudio-S-16k | 72.25 | 8.56 | 28.37 | **19.35** | 6.06 | 19.58 | 0.77 |
| MMAudio-S-16k + NAG (Ours) | **66.32** | **8.69** | **28.78** | 18.51 | **6.63** | **19.96** | **0.73** |

Table A7: Quantitative evaluation of both the composite audio and individual audio tracks generated from step-by-step generation on the Movie Gen Audio Bench.

poor alignment with the text captions or suffer from low quality, such as silence or muffled sound. This may stem from limitations in the base MMAudio model, particularly with handling subtle or rare sounds (e.g., "Carpet rustles gently", "Wings flap gently", "Snowflakes fall silently", "Crowd murmurs quietly"). Even when conditioned only on such text prompts (text-to-audio generation), MMAudio often produces hums, noise, or unnaturally loud sounds, likely due to the scarcity of such audio events in its training data. These mismatches suggest a domain gap between Multi-Caps VGGSound and MMAudio's training distribution. Since NAG only guides generation away from previous outputs, overall quality and text alignment rely heavily on the base TV2A model's capabilities. The effectiveness of our proposed method would likely be more pronounced if the base model supported a broader range of text prompts (i.e., ideally broad enough to match the range supported by LLMs) and could generate more diverse audio outputs, even for the same video input.

**Suboptimal audio mixing process.** In this work, we synthesized composite audio using a simple mixing strategy, summing multiple audio tracks without weighting them, followed by loudness normalization. While effective, it does not account for the natural loudness of each audio track and might be suboptimal. As optimal mixing can differ for each video and audio content, incorporating a generative model to support this process could further enhance the quality of the composite audio. We leave this direction for future work.

## K    ADDITIONAL VISUALIZATIONS

Figure A5 shows additional spectrogram visualizations comparing MMAudio, MMAudio + negative prompting, and MMAudio + NAG. See "demo/index.html" in the supplementary material for more generated samples.

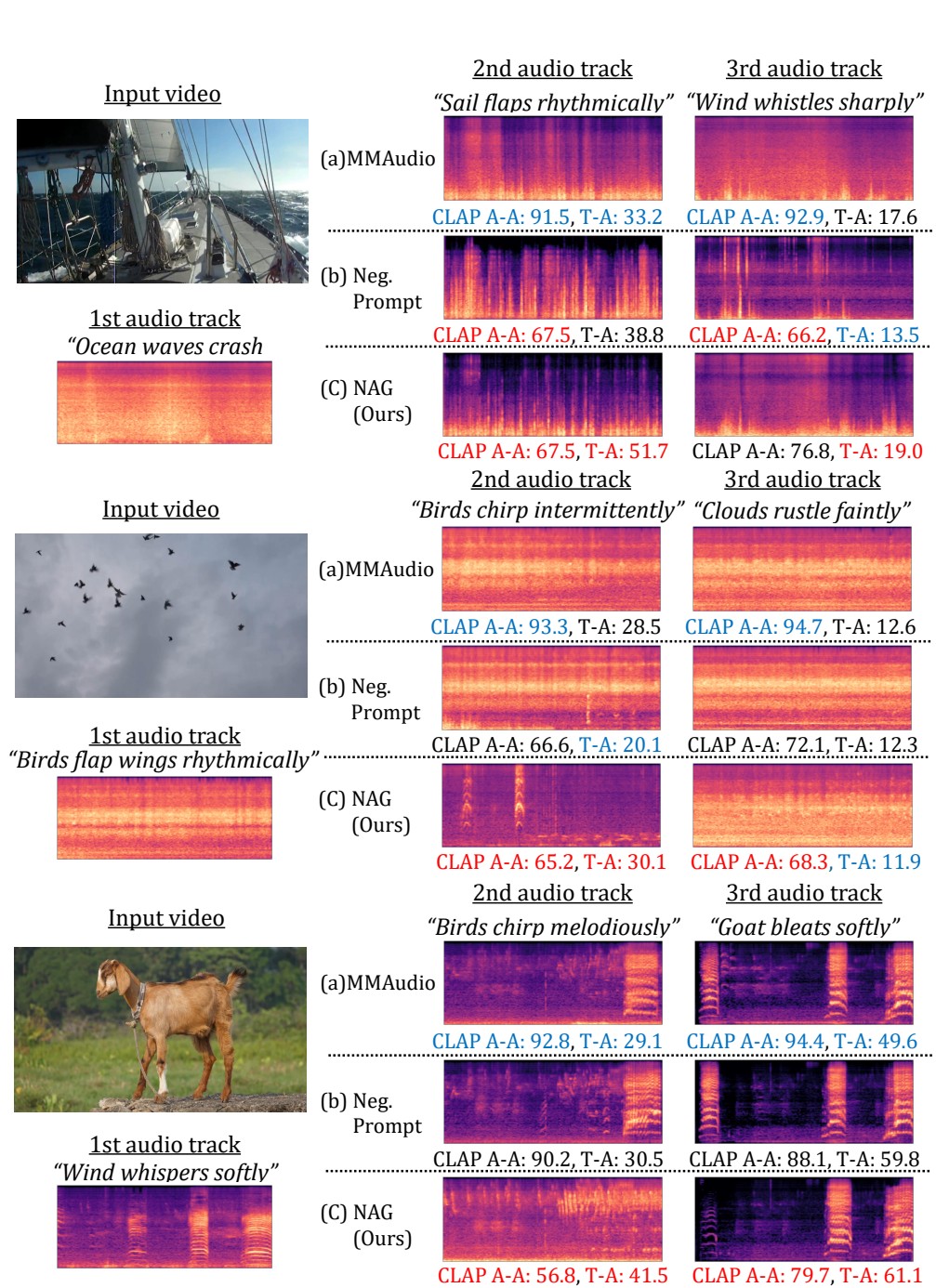

Figure A5: Additional spectrogram visualization. Our proposed method effectively suppresses previously generated sounds in subsequent steps while maintaining high alignment with the text prompts. The best and worst CLAP A-A and T-A scores are highlighted in red and blue, respectively.

