# OpenReview forum: "Step-by-Step Video-to-Audio Synthesis via Negative Audio Guidance"
_ICLR.cc/2026/Conference — Submitted to ICLR 2026_

### Official Review · Reviewer_mo18 · 2025-10-26

**Soundness:** 3
**Presentation:** 3
**Contribution:** 3
**Rating:** 6
**Confidence:** 3

**Summary:**

Inspired by traditional Foley workflows, this paper proposes a step-by-step video-to-audio generation method to overcome controllability limitations of single-pass models. The core method, Negative Audio Guidance (NAG), conditions the generation on previously created audio tracks. This negatively guided process, formulated via concept negation, discourages the duplication of existing sounds and steers the model to incrementally synthesize missing audio events. Critically, the authors introduce the practical contribution of being trainable without costly multi-reference datasets, instead leveraging audio pairs from adjacent segments of the same video.

**Strengths:**

1. To the best of my knowledge, generating audio with separated concepts using audio-based negative guidance is novel in this field. The proposed NAG method is well-justified and, based on the audio separability metrics in Table 2, appears to offer significant improvement. Further, Figure 4 visually demonstrates that the model is capable of the incremental refinement that was challenging for previous V2A models.

2. To overcome the data scarcity problem for this new task, the authors utilized audio pairs from adjacent segments of the same video. This allows the framework to be trained using standard, widely accessible single-reference datasets (e.g., VGGSound).

3. The authors compare their results against strong baselines, showing that step-by-step generation with NAG demonstrates better or comparable results across diverse key metrics, as well as in the user study. Furthermore, the authors constructed the Multi-Caps VGGSound dataset using a VLM to generate multiple, distinct captions per video, which was necessary to evaluate this novel task.

**Weaknesses:**

1. The authors explicitly state this limitation in Appendix G; the overall quality and the ability to generate specific sounds are heavily dependent on the underlying V2A model (MMAudio). NAG is a guidance mechanism that primarily prevents duplication; it seems to struggle with generating more diverse audio outputs or creating subtle or rare sounds if the base model was not adequately trained on them. This creates a hard ceiling on the final audio fidelity.

2. The ablation study in Appendix F (Table A3) reveals that the method's performance is sensitive to the order of generation. This adds a procedural constraint to achieve optimal results, which requires using the ImageBind score as a proxy to determine the generation order.

**Questions:**

1. Some improvements for the composite audio in Table 1 (e.g., IB-score) appear marginal compared to the baseline (MMAudio-S-16k). Please provide the standard deviation for both 'Ours' and 'MMAudio-S-16k' across all objective metrics in Table 1 and Table 2. This would help to ascertain the statistical significance of the observed improvements.

2. As stated in Appendix G, the naive addition is a limitation. Did the authors experiment with any alternative mixing strategies, such as learning the mixing weights or applying techniques like convex optimization, to create the final composite audio?

3. There appear to be discrepancies between the main results for 'Ours' (Tables 1 and 2) and the default setting ($\alpha=4.5, \beta=1.5$) in the ablation study (Table A2). For example, the $FD_{PANNs}$ is 6.47 in Table 1 but 7.32 in Table A2. Please clarify about this discrepancy.

---

> ### Author Response · Authors · 2025-11-22
> **Rebuttal response**
>
> We sincerely thank the reviewer for their thoughtful comments and constructive feedback.
> Below, we address each point in detail.
>
> ## (W1) Dependence on the performance of the base model.
>
> We fully agree with the reviewer that NAG inherits the strengths and limitations of the underlying V2A model.
> This is an inherent property of all guidance-based methods.
> The contribution of our method lies in a generalized guidance framework that is theoretically applicable to any diffusion or flow-matching based generative model (i.e., any model that can estimate scores).
>
> Thus, two axes of improvement remain orthogonal: (i) **Progress in underlying TV2A generative modeling in terms of fidelity, diversity, and coherence with conditions** and (ii) **Improved controllability introduced by NAG**, enabling sequential audio addition without contaminated content.
> In short, future advances in TV2A models will directly translate into higher-quality, controllable multi-track generation under our framework.
>
> ## (W2) Sensitivity on generation order.
>
> We appreciate the reviewer highlighting this point.
>
> For evaluation, we should simulate a human user's workflow, i.e., starting with the most salient sound events and progressively adding missing ones.
> Because such interaction cannot be assumed during automated benchmarking, we use the ImageBind score as a reproducible proxy.
>
> In practice, the generation order is naturally determined by the user: users specify the most important or distinctive sound first, and additional tracks are added only as needed, thereby eliminating the need for model-driven ordering.
> Thus, the order sensitivity observed in the ablation study reflects **evaluation constraints rather than a practical limitation**.
>
> ## (Q1) Standard deviation for both 'Ours' and 'MMAudio-S-16k' in Tables 1 and 2.
>
> We calculated the mean and standard deviation across **five independent runs (five different random seeds)**.
> The results show that improvements are consistent and statistically meaningful.
>
> #### Composite audio (Table 1)
> |Method|$FD_{PANNs}$(↓)|$FD_{VGG}$(↓)|$KL_{PANNs}$(↓)|$KL_{PASST}$(↓)|IS(↑)|IB-Score(↑)|DeSync(↓)|
> |---|:---:|:---:|:---:|:---:|:---:|:---:|:---:|
> |MMAudio-S-16k|7.76 $\pm$ 0.09|1.35 $\pm$ 0.07|2.03 $\pm$ 0.01|1.85 $\pm$ 0.02|10.41 $\pm$ 0.05|28.14 $\pm$ 0.03|0.43 $\pm$ 0.003|
> |Ours|**6.49** $\pm$ 0.08|**0.98** $\pm$ 0.06|**2.01** $\pm$ 0.01|**1.79** $\pm$ 0.03|**10.57** $\pm$ 0.04|**28.62** $\pm$ 0.02|0.43 $\pm$ 0.003|
>
> #### Individual audio track (Table 2)
> |Method|CLAP A-A(↓)|IS(↑)|CLAP T-A(↑)|IB-Score(↑)|
> |---|:---:|:---:|:---:|:---:|
> |MMAudio-S-16k|79.73 $\pm$ 0.02|**12.49** $\pm$ 0.02|28.31 $\pm$ 0.01|**27.74** $\pm$ 0.04|
> |Ours|**71.35** $\pm$ 0.03|12.03 $\pm$ 0.03|**28.89** $\pm$ 0.06|26.62 $\pm$ 0.05|
>
> These results indicate that the improvements are robust to random seeds.
>
> ## (Q2) Advanced mixing strategy.
>
> > Did the authors experiment with any alternative mixing strategies, such as learning the mixing weights or applying techniques like convex optimization, to create the final composite audio?
>
> We did not explore applying convex-optimization-based or learning-based mixing strategies.
>
> Our primary focus is on **controllability in V2A generation**, specifically on generating well-separated audio tracks from previously generated tracks.
> To avoid confounding factors, we intentionally adopted simple summation followed by normalization, which is a transparent, non-heuristic composite strategy.
> A more advanced mixing strategy could obscure the benefit of improved separability and introduce additional learnable components that are orthogonal to our contribution.
>
> We agree that optimized mixing can further improve composite quality and explicitly identify this as an important direction for future work (Appendix I).
>
> ## (Q3) Inconsistency between Tables 1, 2, and A2.
>
> We appreciate the reviewer pointing this out.
>
> (Updated table indexes based on the revised manuscript. Tables A2 and A3 in the original manuscript are now Tables A3 and A5, respectively.)
> The results in Table A3 correspond to the **Random Order** setting, which is not used in the main experiments.
> Table A5 includes both Random and ImageBind-based ordering, and the numbers in Table A3 match the Random configuration.
> Note that the same order is used across all settings, and the comparison among guidance scales is fair.
>
> Thus, there is no conflict:
> - Tables 1 and 2: ImabeBind-based ordering.
> - Table A3: Random ordering.
> - Table A5: comparison of generation order.
>
> We have clarified this distinction in the revision.

---

### Official Review · Reviewer_d7qG · 2025-10-31

**Soundness:** 3
**Presentation:** 3
**Contribution:** 2
**Rating:** 4
**Confidence:** 4

**Summary:**

This paper introduces a step-by-step video-to-audio (V2A) generation framework that mimics the Foley workflow, where missing sound events are incrementally added.
To prevent duplication of already-generated sounds, the authors propose Negative Audio Guidance (NAG) that pushes new audio away from previously generated audio features.
The method fine-tunes MMAudio with a ControlNet.
For evaluation, they build Multi-Caps VGGSound, a dataset with multiple captions per video to represent multiple sound events.
Experiments show improved audio separability across generated tracks and slightly better audio–video alignment compared to MMAudio and negative-prompt baselines.

**Strengths:**

1.	First to explicitly tackle incremental, Foley-style video-to-audio generation via negative conditioning.
2.	The method is compatible with existing flow matching diffusion frameworks.
3.	The authors propose a new dataset (Multi-Caps VGGSound).

**Weaknesses:**

1.	The number of generation steps is fixed to 5 for all 8-second clips, regardless of actual sound density. Adaptive step selection or early stopping could make the framework more efficient and realistic.
2.	The gain over the MMAudio baseline is marginal. Since MMAudio already produces high-quality, semantically aligned sounds, it remains unclear why a multi-step process is needed beyond marginal improvements in separability. More compelling examples (e.g., complex or dense soundscapes) would strengthen. Or a direct comparison with fine-tuned MMAudio trained on the multi-caption dataset would make the advantage clearer.

**Questions:**

1.	In the demos, temporal alignment appears less critical than semantic correctness (e.g., when a pigeon appears, the model simply adds wing-flapping sounds). This is also reflected in Figure 5 and Table 1, where temporal alignment shows the least improvement. Have the authors considered how it might be improved?
2.	Since Qwen cannot handle audio input, have the authors thought of other evaluation benchmarks (e.g., VGGSounder (ICCV 2025), which include multiple audio/video labels per clip)?
3.	Given that the proposed method requires roughly 10 seconds of additional generation time compared to MMAudio, do the authors believe this trade-off is worthwhile in practice?
4.	The paper could discuss and compare with recent ControlNet-style or additive generative approaches such as:
- SonicVisionLM:  Playing Sound with Vision Language Models (CVPR 2024)
- Read, watch and scream! sound generation from text and video (AAAI 2025)

---

> ### Author Response · Authors · 2025-11-22
> **Rebuttal response (1/3)**
>
> We appreciate the reviewer’s thoughtful feedback.
>
> Before addressing each point, we would like to clarify the primary goal of our work.
> Our focus is to **enable sequential V2A generation for controllability**, inspired by the Foley workflow.
> Unlike one-step V2A methods that aim for fully automatic end-to-end generation, Foley-style applications require **fine-grained control**, where users selectively add or regenerate specific sound events.
> The ultimate goal is not only to produce audio that matches the ground-truth recording, but also to support the creation of **immersive sound content tailored to user intent**.
>
> In this setting, **audio separability becomes essential**, because well-separated tracks allow users to freely regenerate or add desired audio events without affecting others.
> Thus, our primary contribution is to improve controllability and separability while maintaining audio quality and video–audio alignment, rather than merely maximizing composite-audio performance.
> This focus is clearly stated in the Introduction and reflected in our evaluation design.
>
> Below, we address each point in detail.

---

> ### Author Response · Authors · 2025-11-22
> **Rebuttal response (2/3)**
>
> ## (W1) Fixed number of generation steps.
>
> We agree that the number of steps should be flexible depending on the video application.
> We fixed the number of steps solely for evaluation consistency.
> In practice, the number of steps is naturally determined by user interaction, as users selectively add missing, desired sound events.
>
>
> ## (W2 / Q3) Why is a multi-step process needed?
>
> The primary reason is that we want to **improve the controllability** in the V2A generation process.
> Adding missing sound events using existing fully end-to-end models is challenging since those models duplicate the regeneration of already generated sound events.
> This process is unintuitive, as a user needs to regenerate entire audio tracks from scratch even though only a specific audio event needs to be added or revised.
>
>
> > A direct comparison with fine-tuned MMAudio trained on the multi-caption dataset would make the advantage clearer.
>
> We cannot directly fine-tune MMAudio, since there is no publicly available multi-reference audiovisual dataset.
> Constructing such a dataset at scale and with high quality is also infeasible, as visually-guided audio source separation remains a challenging open problem (as we discussed in Section 3.2).
> Our approach is specifically designed to avoid the need for such multi-reference datasets, which are highly costly to construct and remain scarce in existing V2A research.
> Our method provides a practical solution to this dataset scarcity; it achieves sequential V2A generation **only using a standard single-reference audiovisual dataset**.
> This point is a primary contribution of this paper.
>
> > Do the authors believe this trade-off is worthwhile in practice?
>
> Yes.
> As we stated above, our primary focus is to achieve better controllability in V2A generation, similar to the Foley workflow.
> To achieve this, a sequential generation approach is essential, enabling a user to add missing sound events step-by-step to synthesize immersive sound content.
>
> We argue that **one-step generation inherently misses some concepts even though they are specified in a caption**.
> To clarify whether one-step results successfully contain multiple events, we apply AudioSep [1] to single-track outputs from MMAudio-S-16k with fused captions.
> We use the official AudioSep model with its default configuration and the same captions for separation as in our step-by-step generation.
>
> |method|CLAP A-A(↓)|IS(↑)|CLAP T-A(↑)|IB-Score(↑)|
> |---|:---:|:---:|:---:|:---:|
> |One-step + postprocessing|79.11|9.04|23.42|21.96|
> |Ours|**71.38**|**12.01**|**28.91**|**26.67**|
>
> The separated tracks exhibit worse text-video alignment (CLAP T-A and IB-Score), indicating that the one-step results often fail to capture all concepts from the fused captions.
> Higher (worse) CLAP A-A similarity also highlights this failure, as missing concepts lead to silent or noisy separated tracks.
> Based on this analysis, a sequential approach is a practical solution, allowing incremental addition of missing sound events to improve controllability.
> This justifies the needs of the proposed sequential generation.
>
> [1] "Separate What You Describe: Language-Queried Audio Source Separation", 2023
>
> ## (Q1) Improvement on temporal alignment.
>
> > Have the authors considered how it might be improved?
>
> We did **not** specifically focus on further improving temporal alignment in this work.
> Instead, we introduce NAG to **improve the separability** of each generated audio track while maintaining the base model's temporal alignment performance.
> Table 1 and Fig. 5 show a clear improvement in audio separability while achieving on-par or marginally better temporal alignment scores, illustrating that our method successfully guides the base model to generate well-separated multiple audio tracks without sacrificing temporal alignment.

---

> > ### Author Response · Authors · 2025-11-22
> > **Rebuttal response (3/3)**
> >
> > ## (Q2) Rationale of using Qwen2.5-VL for dataset construction.
> >
> > > Have the authors thought of other evaluation benchmarks (e.g., VGGSounder (ICCV 2025), which include multiple audio/video labels per clip)?
> >
> > We did **not** consider this direction because our evaluation setup targets Foley-style scenarios.
> > We construct a dataset for evaluating a Foley-like V2A generation process, in which a user begins with a silent video and adds plausible sound events induced solely from visual cues.
> > As our goal is not to reconstruct ground-truth audio, the generated captions do not need to align with the actual sound in the video.
> > To cover diverse audio events and avoid biasing the captions by the ground-truth audio, we adopted Qwen2.5-VL for dataset construction.
> >
> > To demonstrate the generalizability of the dataset construction pipeline (and the proposed method),  we conducted additional experiments on the Movie Gen Audio Bench [2] dataset.
> > This dataset consists of generated videos produced by Movie Gen and does not contain ground-truth audio tracks (i.e., all videos are silent).
> > Our dataset construction process can be adopted to such a scenario, which a captioning system relying on both audio and video (e.g., VGGSounder) does not cover.
> >
> > #### Composite audio results
> > |(Movie Gen Audio Bench)|IS(↑)|IB-Score(↑)|DeSync(↓)|
> > |---|:---:|:---:|:---:|
> > |MMAudio-S-16k|6.06|19.58|0.77|
> > |Ours|**6.63**|**19.96**|**0.73**|
> >
> > #### Individual audio track results
> > |(Movie Gen Audio Bench)|CLAP A-A(↓)|IS(↑)|CLAP T-A(↑)|IB-Score(↑)|
> > |---|:---:|:---:|:---:|:---:|
> > |MMAudio-S-16k|72.75|8.56|28.37|**19.35**|
> > |Ours|**66.32**|**8.69**|**28.78**|18.51|
> >
> >
> > [2] Movie gen: A cast of media foundation models, 2024
> >
> > We also note that VGGSounder was accepted to ICCV 2025, which takes place after the ICLR 2026 submission deadline, and therefore consider it concurrent work.
> >
> > ## (Q4) Discussion with recent ControlNet-style or additive generation approaches.
> >
> > We appreciate the reviewer for pointing out these related works.
> >
> > **ReWaS** [3] proposes a video-and-text-to-audio model that injects continuous audio energy predicted from video into a text-to-audio generator via ControlNet, thereby improving temporal coherence.
> > While both our method and ReWaS incorporate additional conditioning into an existing audio generator, ReWaS does not aim to support a sequential V2A workflow.
> > Their framework focuses on enhancing temporal alignment in a single-step generation process.
> > In contrast, our work focuses on **enabling controllable, step-by-step V2A generation** in which the generated audio track can be incrementally added without affecting previously generated tracks.
> >
> > **SonicVisionLM** [4] introduces a timestamp-conditioned video-and-text-to-audio generation model.
> > They first convert the video into text descriptions, then generate audio using a text-to-audio model conditioned on a user- or video-provided timestamp.
> > This design allows additive audio generation, since the model is inherently a text-to-audio model and therefore does not suffer from audio contamination from visual conditioning.
> >
> > However, this approach has several fundamental limitations:
> > - **Loss of fine-grained audiovisual synchronization**: Because the video is converted to text, the model cannot leverage the detailed semantic and temporal cues captured by multimodal V2A models such as Multi-Foley, MMAudio, and Kling-Foley. This limitation makes it challenging for SonicVisionLM to achieve the same level of fine-grained audiovisual synchronization targeted by recent multimodal V2A models.
> > - **Limited support for complex or subtle sound events**: Temporal alignment relies on explicit timestamp input, and the model mainly supports clearly visible actions. Events with ambiguous timing, indirect causality, or weak visual cues are challenging to capture.
> > - **Requirement of a specialized dataset.**: Training requires video-audio-text-timestamp-aligned data, which is expensive to construct at scale.
> >
> > In contrast, our method can be applied to any multimodally trained TV2A model that supports video conditioning (e.g., Multi-Foley, MMAudio, etc.).
> > SonicVisionLM, in its current form, is built on a T2A backbone and would face the same contamination issue if extended to such TV2A settings.
> > By introducing NAG, we explicitly address the issue of **audio contamination that naturally arises when conditioning on visual features**, enabling these multimodal models to be extended to **additive and controllable audio generation** without relying on specialized timestamp labels or text-only conditioning.
> >
> > We have incorporated these discussions into the revision.
> >
> > [3] Read, watch, and scream! Sound generation from text and video, AAAI 2024
> > [4] SonicVisionLM: Playing sound with Vision Language Models, CVPR 2025

---

> > > ### Comment · Reviewer_d7qG · 2025-11-26
> > >
> > > I appreciate the authors’ clarifications and additional experiments. I have a few further questions and concerns.
> > >
> > > **(W1) Number of generation steps**
> > >
> > > The authors agree that the number of steps should ideally be flexible and depend on the sound density of each video.
> > > If so, is there any ablation or analysis showing how performance changes with different numbers of steps? Moreover, it is unclear whether a multi-step design is necessary in cases with only a small number of sound events. In such scenarios, independent MMAudio generation might already provide sufficient controllability and separability.
> > >
> > > **(W2 / Q3) Contribution relative to MMAudio**
> > >
> > > Authors emphasize controllability and separability as the main contributions in this rebuttal. However, many existing Foley-style V2A methods already incorporate text inputs, which also provide explicit control over which sound events should be generated. Given this, in my opinion, the most directly relevant baseline would be independent MMAudio generation per caption (rather than the one-step one caption setting), which corresponds to the “MMAudio in independent generation” in Table 1.
> > >
> > > However, as written in my initial review, the improvements over independent MMAudio appear to be very marginal. Given this, I am still unsure whether the proposed sequential Foley-style framework provides a sufficiently strong standalone contribution. Is there any additional baseline beyond MMAudio where the method shows a clearer or more generalizable advantage?
> > >
> > > I will also consider other reviewers’ perspectives.

---

> ### Author Response · Authors · 2025-12-01
>
> We thank the reviewer for asking additional questions and for giving us the opportunity to address the reviewer's concerns.
> Although we do not have an opportunity to receive further feedback, we would like to address the reviewer's points as clearly as possible.
>
> ## (W1) Number of generation steps
> We appreciate the reviewer's suggestion to analyze performance differences with respect to the number of sound events.
> To examine whether a multi-step design is necessary even in cases with only a few events, we conducted additional experiments on the VGGSounder dataset.
>
> The results are shown in Section H and Figure A4.
> Overall, our method consistently improves the separability of individual audio tracks and the quality of the composite audio across different numbers of captions (including cases with only two sound events), while maintaining both the audio quality and text fidelity of the individual audio tracks.
> Please refer to the revised manuscript for details.
>
> ## (W2 / Q3) Contribution relative to MMAudio
>
> > Many existing Foley-style V2A methods already incorporate text inputs, which also provide explicit control over which sound events should be generated.
>
> We agree that existing V2A models use text conditioning.
> However, as discussed in the introduction and demonstrated in Figure 4, relying solely on text control (with a fixed video input) often yields unsatisfactory results: prominent visual events leak into the generated audio even when they are not described in the text.
> This imperfect disentanglement between video and text motivates our approach, which improves controllability using the generated audio itself.
>
> > The most directly relevant baseline would be independent MMAudio generation per caption (rather than the one-step one caption setting)
>
> The reviewer is correct.
> Independent MMAudio generation per caption is our primary baseline.
> Accordingly, we compare the proposed method against this baseline comprehensively through objective and subjective evaluations.
> Beyond Table 1, Tables 2, A1-A2, and A5-7, and Figures 4-5 and A4 all report comparisons against independent MMAudio generation.
>
> > The improvements over independent MMAudio appear to be very marginal.
>
> We understand the reviewer's concern regarding the significance of the improvements.
> To make this clear, we provide both quantitative and subjective evaluations, which consistently show a meaningful gain with our method.
> We also added statistical analysis for both 'Ours' and 'MMAudio-S-16k' in Tables 1 and 2.
> Please refer to our response to Reviewer mo18 for details.
>
> Finally, even if one considers the composite audio quality improvement marginal, the contribution remains valid: our method reliably improves the separability of the individual audio tracks without degrading the composite audio quality.
> As discussed in the Introduction, this separability improvement is crucial for controllable, Foley-style video-to-audio generation.

---

> > ### Author Response · Authors · 2025-12-01
> >
> > Once again, we sincerely thank Reviewer d7qG for providing additional feedback on our rebuttal.
> > Their thoughtful comments have been constructive in further improving our manuscript.

---

### Official Review · Reviewer_RTJi · 2025-10-31

**Soundness:** 2
**Presentation:** 3
**Contribution:** 2
**Rating:** 2
**Confidence:** 4

**Summary:**

This paper proposes a step-by-step video-to-audio(V2A) generation method that avoids duplicating previously generated sounds through Negative Audio Guidance (NAG). Additionally, the authors construct a new audiovisual benchmark dataset, named Multi-Caps VGGSound, which supports evaluation of compositional and incremental sound generation in V2A tasks. Experiments on the proposed benchmark across diverse aspects show that the proposed method improves audio separability, text fidelity, and overall audio quality compared to strong baselines. In summary, this work offers a practical framework for controllable and compositional Foley-like sound generation.

**Strengths:**

Unlike conventional V2A models that generate an entire audio tracks in a single pass, this paper introduces an incremental refinement framework inspired by real-world Foley workflows. The authors provide a strong theoretical foundation for the effectiveness of Negative Audio Guidance (NAG) by leveraging well-established techniques such as flow matching and classifier-free guidance, and empirically validate it thorough experiments. The proposed framework consistently outperforms state-of-the-art baselines across multiple metrics, including audio quality, semantic and temporal alignment.
The paper is clearly written and fairly easy to follow.

**Weaknesses:**

1.	While the proposed NAG framework introduces a promising way to suppress previously generated audio, it is unclear whether the model can effectively disentangle composited audio to understand which events have already been generated and should be suppressed in subsequent steps.

2.	The experimental evaluation relies solely on the VGGSound-based constructed dataset (Multi-Caps VGGSound), which may limit the generalizability of the proposed method. Although the authors proposed a method to construct evaluation benchmarks by generating multiple captions using a vision-language model, this procedure could have been applied to other datasets as well (e.g., AudioCaps) to better assess the robustness and applicability of the method in diverse audiovisual contexts.

3.	While the method is compared with several variants of MMAudio and negative prompting strategies, the evaluation lacks diversity in terms of architecture and training paradigms. Demonstrating the application of NAG to other types of V2A models would strengthen the claim of its generalizability beyond a single base model.

4.	According to the paper, the user study for composite audio was conducted with only 10 participants. This limited scale raises concerns about the statistical significance of the results. A larger and more diverse user study would provide stronger evidence of the method’s perceptual effectiveness.

**Questions:**

-	How effectively can the model distinguish individual audio events from the composited audio used in the NAG condition, especially in cases with overlapping or complex sound mixtures? Is there any analysis on the model’s capacity for selective suppression in such scenarios?

-	Given that the authors construct the evaluation set (Multi-Caps VGGSound) by generating multiple captions using a vision-language model, have authors considered applying the same procedure to other standard audiovisual datasets such as AudioCaps to better validate the generalizability and robustness of their method?

-	Is there any reason authors used Qwen2.5-VL model instead of other vision language model?

-	Have the authors explored integrating NAG with other V2A architectures beyond MMAudio, particularly those that support classifier-free guidance, to better validate the generalizability of the proposed method?

-	To improve the reliability of the user study for the composite audio, would the authors consider conducting a larger-scale user study and providing more detailed statistical analysis?

---

> ### Author Response · Authors · 2025-11-22
> **Rebuttal response (1/2)**
>
> We sincerely appreciate the reviewer's constructive feedback.
> Below, we address each point in detail.
>
> ## (W1 / Q1) Disentanglement of NAG conditioning.
>
> > Is there any analysis on the model’s capacity for selective suppression in such scenarios?
>
> Yes.
> We provide quantitative evidence of selective suppression in Table 2, using the CLAP A-A similarity metric across generated audio tracks for the same video.
>
> **NAG successfully reduces the similarities between audio tracks**, indicating that it suppresses previously generated content more effectively than baselines.
> This analysis covers **all 10 track pairs** for each video (10 combinations from 5 generated tracks), covering cases where multiple sound events are fused and used as a negative condition.
>
> Below, we clarify the evaluation setup:
> For each video, we generate 5 audio tracks and compute distances of CLAP audio embeddings for all audio track pairs $\{(x^{(i)},x^{(j)})\}_{i < j}$, and report the mean over all videos.
> Please see Section 5.1 ("Task setup" and "Evaluation metrics") for more details.
>
> The results demonstrate that NAG improves separability (79.55 → 71.38, lower is better) while maintaining overall quality.
> Furthermore, the demo in the supplementary material confirms that the 5th track does not contain audio from 1st to 4th tracks, illustrating **successful multi-event suppression**.
>
> ## (W2 / Q2) Other datasets (AudioCaps & MovieGenBench).
>
> > Have authors considered applying the same procedure to other standard audiovisual datasets such as AudioCaps to better validate the generalizability and robustness of their method?
>
> Yes.
> To validate the generalizability across datasets, we additionally evaluate our method on the AudioCaps [1] and the Movie Gen Audio Bench [2].
> We follow the same caption-generation procedure as Multi-Caps VGGSound, applying Qwen2.5-VL to generate multiple event-level captions.
>
> ### Composite audio results
> |(AudioCaps)|IS(↑)|IB-Score(↑)|DeSync(↓)|
> |---|:---:|:---:|:---:|
> |MMAudio-S-16k|6.81|25.73|0.56|
> |Ours|**7.08**|**26.36**|**0.54**|
>
> |(Movie Gen Audio Bench)|IS(↑)|IB-Score(↑)|DeSync(↓)|
> |---|:---:|:---:|:---:|
> |MMAudio-S-16k|6.06|19.58|0.77|
> |Ours|**6.63**|**19.96**|**0.73**|
>
> ### Individual audio track results
> |(AudioCaps)|CLAP A-A(↓)|IS(↑)|CLAP T-A(↑)|IB-Score(↑)|
> |---|:---:|:---:|:---:|:---:|
> |MMAudio-S-16k|68.89|**9.65**|29.09|**24.91**|
> |Ours|**60.43**|9.29|**29.25**|23.46|
>
> |(Movie Gen Audio Bench)|CLAP A-A(↓)|IS(↑)|CLAP T-A(↑)|IB-Score(↑)|
> |---|:---:|:---:|:---:|:---:|
> |MMAudio-S-16k|72.75|8.56|28.37|**19.35**|
> |Ours|**66.32**|**8.69**|**28.78**|18.51|
>
> These results align with Tables 1 and 2, demonstrating that our method consistently improves audio separability while preserving overall quality across datasets.
> We have updated the manuscript to include these results.
>
> [1] Audiocaps: Generating captions for audios in the wild, NAACL-HLT, 2019
> [2] Movie gen: A cast of media foundation models, 2024
>
> ## (Q3) Reason why Qwen2.5-VL.
> > Is there any reason authors used Qwen2.5-VL model instead of other vision language model?
>
> Yes.
> We chose Qwen2.5-VL for its accessibility and high-quality, consistent captioning.
>
> In preliminary tests, we manually compared Qwen2.5-VL with LLaVA [3] and PPLLaVA [4].
> LLaVA and PPLLaVA frequently produced hallucinations (e.g., mentioning a cello and a contrabass despite the video containing only a violinist).
> Qwen2.5-VL generated captions that remained consistent with visible entities and plausible background sound events, making it more suitable for constructing reliable evaluation prompts.
>
> [3] Visual Instruction Tuning, NeurIPS, 2023
> [4] PPLLaVA: Varied Video Sequence Understanding With Prompt Guidance, 2024

---

> ### Author Response · Authors · 2025-11-22
> **Rebuttal response 2/2**
>
> ## (W3 / Q4) Applicability to other V2A architectures.
> We agree that evaluating NAG on additional architectures would further strengthen the claim.
> However, **applying NAG requires (i) text- and video-conditioned generation and (ii) access to both training and inference pipelines**, and, to the best of our knowledge, no existing models satisfy these premises.
> - Diff-Foley, V2A-Mapper, and Frieren do not support text conditioning and are incompatible with our sequential, text-driven workflow.
> - MultiFoley and Kling-Foley support both video and text conditions but are closed-source, making integration and fair comparison infeasible.
>
> Currently, only **MMAudio** provides complete training and inference details, enabling a rigorous implementation of our method and a fair comparison with the baseline.
>
>
> ## (W4 / Q5) Reliability of the user study.
>
> We understand the reviewer's concern.
> Below, we contextualize the scale and provide statistical analysis.
>
> ### Comparison with prior work in terms of the scale
> We summarize the number of evaluators and responses reported in prior work and ours.
> - Frieren: 6 evaluators, 900 responses.
> - Multi-Foley: 20 evaluators, 400 responses.
> - MMAudio: 23 evaluators, 920 responses.
> - **Ours: 10 evaluators, 450 responses.**
>
> Our scale is comparable, and not the smallest among existing work.
>
> ### Statistical reliability
> Following prior work (Frieren, Multi-Foley), we compute a **95% confidence interval (CI)** using the Wilson score interval (for preferences) and the standard error (for ratings).
>
> #### 95% CI for composite audio preferences
>
> ||Audio quality(↑)|semantic alignment(↑)|Temporal alignment(↑)|
> |---|:---:|:---:|:---:|
> |Win rate|71.36 $\pm$ 7.71 |76.00 $\pm$ 7.62|61.14 $\pm$ 7.85|
>
>
> #### 95% CI for rating scores on individual audio tracks
>
> |method|Separability(↑)|Audio quality(↑)|Text fidelity(↑)|
> |---|:---:|:---:|:---:|
> |MMAudio-S-16k|2.24 $\pm$ 0.15 |2.89 $\pm$ 0.14|2.42 $\pm$ 0.18|
> |Ours|3.35 $\pm$ 0.15|3.30 $\pm$ 0.14|3.12 $\pm$ 0.18|
>
> The confidence intervals show that our improvements are statistically significant relative to the baseline.
> We have incorporated this analysis into the revision to clarify the user study's reliability.

---

### Official Review · Reviewer_yhaW · 2025-11-01

**Soundness:** 2
**Presentation:** 3
**Contribution:** 3
**Rating:** 6
**Confidence:** 3

**Summary:**

The paper introduces a method to sequentially generate multiple audio tracks from video. To achieve this, the authors derive a technique called "Negative Audio Guidance" (NAG). During inference, NAG uses a reference audio-based model to prevent the generation of sounds that already exist in the audio track. The effectiveness of this method is demonstrated through quantitative and qualitative results on the VGGSound dataset.

**Strengths:**

1. This paper is well-written and it is easy to follow.

2. The step-by-step generation process guided by negative audio is a novel and intuitive approach to audio synthesis, offering more control over the final output.

3. The method's ability to train without requiring complex, multi-reference datasets makes it more practical and accessible for researchers.

4. The paper provides a comprehensive evaluation, including both objective metrics and a subjective user study, to validate the effectiveness of the proposed method.

**Weaknesses:**

1. The motivation of generating multiple audio tracks step by step is not well-verified. The authors should compare such an step-by-step method with those methods with single-step inference with postprocessing (audio tracks decomposition).

2. The dataset used for evaluation ("Multi-Caps VGGSound") was created using a vision-language model without access to the original audio, which could introduce a discrepancy between the text captions and the actual sound events. The authors should provide detailed quality analysis of such a dataest.

3. The central idea, "Negative Audio Guidance" (NAG), is presented as a key contribution. However, this is functionally an application of negative prompting or classifier-free guidance, techniques that are well-established in the broader field of conditional diffusion models, particularly in the image domain. The paper's novelty lies in applying this to a sequential audio generation framework, but it would be stronger if it were positioned as an application and adaptation of existing principles rather than the invention of a fundamentally new guidance method. The contribution is more in the "how" (the sequential process) than the "what" (the negative guidance itself).

4. The paper's stated goal is to mimic the Foley process to create a complete, realistic audio track. However, the evaluation primarily measures the separability and individual quality of the generated sounds (stems), not the coherence and realism of the final composite audio. The simple summation used for mixing is a critical point of failure for realism. A realistic soundscape is not just a sum of its parts; it involves complex interactions, occlusion, and environmental acoustics. The current evaluation does not adequately measure whether the final output sounds like a plausible, unified audio scene.

**Questions:**

1. Could you elaborate on the relationship between NAG and existing negative prompting techniques in diffusion models? A more explicit positioning of your work within this broader context would help clarify whether the primary contribution is the guidance mechanism itself or its application within your proposed sequential generation framework.

2. We suggest the authors also compare single audio-track generation with completed audio track generation with post-processing (e.g., audio tracks decomposition) to evaluate the performance of a single audio track.

---

> ### Author Response · Authors · 2025-11-22
> **Rebuttal response (1/2)**
>
> We sincerely appreciate the reviewer's time and thoughtful feedback.
> Below, we address each point in detail.
>
> ## (W1) Motivation for step-by-step generation rather than single-step generation with post-processing.
>
> We adopt step-by-step generation to more reliably cover the user-specified audio events as comprehensively as possible.
>
> As the reviewer noted, single-step generation followed by post-processing would be a straightforward alternative.
> However, this approach fails for two reasons: 1) **single-step generation often misses concepts** when the text prompt contains multiple audio events, and 2) **audio source separation degrades quality**, producing artifacts and unstable separated tracks.
> It also complicates user interaction: if a user wants to revise only a specific sound event, they must re-run both the generator and a separation model, which is unintuitive and fragile.
>
> In contrast, our step-by-step design explicitly ensures that the model focuses on only the missing sound event specified at each step.
> Each step isolates one concept, making it far less likely that multiple events interfere with each other, which is the primary reason for concept omission in the single-step pipeline.
> This design also provides an intuitive user interface that lets a user focus on a specific audio event at each step.
> As we show in our comparison with one-step generation plus post-processing (W1/Q2), this design choice leads to clear improvements in both alignment and quality metrics.
>
> ## (W1 / Q2) Comparison against single audio-track generation with post-processing.
> To clarify the limitations of one-step generation with post-processing, we evaluated separated audio tracks obtained by applying AudioSep [1] to single-track outputs from MMAudio-S-16k with fused captions (the composite sound results are shown in Table 1 in the paper).
> We use the official AudioSep model with its default configuration and the same captions for separation as in our step-by-step approach.
> The following shows the comparison between our step-by-step method and one-step generation with post-processing.
>
> |method|CLAP A-A(↓)|IS(↑)|CLAP T-A(↑)|IB-Score(↑)|
> |---|:---:|:---:|:---:|:---:|
> |One-step + postprocessing|79.11|9.04|23.42|21.96|
> |Ours|**71.38**|**12.01**|**28.91**|**26.67**|
>
> The separated tracks exhibit worse text-audio and video-audio alignment (CLAP T-A and IB-Score) because the one-step results often fail to capture all concepts from the fused captions.
> Missing concepts lead to silent or noisy separated tracks, hurting both audio separability (CLAP A-A) and overall quality.
> These results demonstrate that one-step generation with post-processing is a suboptimal strategy for achieving multiple audio tracks, whereas our sequential method successfully mitigates these issues.
>
> We have included the above quantitative results and discussion in the revision to make the limitations of the one-step + post-processing approach explicit.
>
> [1] "Separate What You Describe: Language-Queried Audio Source Separation", 2023
>
> ## (W2) Using a vision-language model for constructing the Multi-Caps VGGSound dataset.
>
> We argue that **evaluation prompts do not need to match the ground-truth audio** in the original video, because our goal is not to reproduce the recorded soundtrack but to support controllable, Foley-style sound creation.
> In Foley workflows, sound designers intentionally create sound events based on visual cues: they do not always aim to recover the original audio but to craft immersive soundscapes aligned with their intent.
> Therefore, in evaluating controllable V2A generation, **evaluation prompts should represent diverse, plausible sound events inferred from the visual cues**, rather than the specific sounds captured in the original recording.
>
> Our method aims to provide fine-grained controllability, enabling a creator to add sound events selectively.
> To evaluate this capability, it is essential to test whether the model can faithfully generate each intended sound event as an independent, well-separated audio track.
> To this end, we use a vision–language model to produce multiple plausible descriptions of sound events inferred from each video, serving a realistic setting for user-specified Foley prompts.
>
> We acknowledge that using VLM-generated captions introduces some noise, as not all descriptions perfectly match the actual audio.
> However, we manually inspected a subset and confirmed that the captions are visually plausible.
> Importantly, all compared methods are evaluated using the same VLM-generated captions, so the evaluation remains fair across methods even if the captions contain some noise.
>
> We view this setup as a proxy for Foley-style use cases where the original recording does not fix the target audio.
> We have clarified this limitation and the intended evaluation setting in the manuscript.

---

> ### Author Response · Authors · 2025-11-22
> **Rebuttal response (2/2)**
>
> ## (W3 / Q1) Relationship between NAG and existing negative prompting.
>
> We agree with the reviewer's point.
> The key contribution of this paper is to enable sequential V2A generation using only a single-reference audiovisual dataset via a negative guidance framework.
> This avoids the need for a multi-reference audiovisual dataset (required for direct modeling), which is hard to obtain at scale.
>
> We formulate sequential V2A generation as the problem of modeling $p(x|V, C_p, C_n)$, where x is audio, V is video, and C_p and C_n denote positive and negative concepts, respectively.
> Instead of training this distribution directly, we decompose it into a chain of distributions (Eq. (5)) and apply a generalized guidance framework for multiple conditions (Du et al., 2020; Liu et al., 2022).
> This yields Eq. (7) and allows us to avoid multi-reference training. (The mechanism is conceptually similar to negative prompting, as we can use the same text-conditional generator to estimate a negatively conditioned score.)
>
> Note that classifier-free guidance (CFG) and negative prompting are special cases of this guidance formulation.
> Using the fused previous-step prompt as a negative prompt for $C_n$ yields $u(x) = u(x) + \alpha(u(x, V, C_p) - u(x)) + \beta(u(x, V, C_p) - u(x, V, C_n))$ (which corresnponds to Eq.(7)).
> Here, $\beta=0$ with arbitrary $\alpha$ recovers CFG, and $\alpha=1$ with arbitrary $\beta$ recovers a typical negative prompting.
> In this sense, NAG is positioned as a new instantiation of guidance for V2A generation, explicitly tailored for the sequential process.
>
> The essential difference between NAG and negative prompting lies in representational capacity.
> Negative prompting requires a textual description that fully expresses the previously generated sound, but converting audio to text always loses information (e.g., timing, style, amience).
> In contrast, NAG conditions directly on the **actual generated audio**, enabling more precise negative guidance without explicit audio-text conversion.
>
> Concretely, what is new in NAG is not the guidance formula itself, but the way we exploit it to (i) use previously generated audio as a negative condition and (ii) enable sequential V2A generation using only single-reference training data.
>
>
> ## (W4) Goal of this paper and evaluation on composite sound.
>
> Our goal is to achieve **greater controllability in V2A generation**, inspired by the Foley workflow, by enabling the sequential addition of missing sound events via text prompts.
> Accordingly, we focus on improving **audio separability** and **text fidelity** for each stem while preserving audio quality and video alignment.
> Since each generated stem can be freely mixed according to user preference, producing high-quality, well-separated sound events is inherently valuable.
>
> In addition to separability and individual stem quality, we evaluate the composite sound in terms of audio realism and semantic coherence with the video (Table 1).
> Although the separability of each generated track is our primary contribution, we also include composite sound evaluation using the most transparent baseline: simple summation followed by normalization.
> We deliberately avoid introducing additional heuristic mixing choices that could confound the comparison across methods.
> We acknowledge that the mixing strategy significantly affects composite quality, especially in the end-to-end V2A pipeline, and explicitly list this as a limitation and a future direction (Sec. I).
>
> To clarify our motivation, we revised the abstract sentence to:
> > "Inspired by traditional Foley workflows, our approach aims to provide better controllability by enabling incremental generation of desired sound, thus enabling users to produce multiple sound events induced by a video comprehensively.".
>
> We thank the reviewer for pointing out the need to strengthen this explanation.

---

### Author Response · Authors · 2025-11-22
**To all reviewers**

First, we would like to express our sincere gratitude to all reviewers for taking the time to provide invaluable feedback on our manuscript.
We have carefully considered all comments and questions and have provided detailed responses to each reviewer.

In response to the feedback, we have updated our manuscript (all revisions are marked in red).
The main modifications are summarized as follows:
- We revised the abstract to better **clarify the goal** of this paper.
- We added **two important related works (ReWaS and SonicVisionLM)** in Section 2 and discussed the differences and limitations relative to our approach.
- To **clarify the statistical significance of the user study**, We added confidence-interval analysis in Tables A1 and A2 .
- To **discuss the advantages and limitations of using VLM for dataset construction**, we added a paragraph in Section C.
- To **clarify the distinction between Tables 1, 2, and A3**, we expanded the description of Table A3.
- To **elaborate on the motivation and significance of step-by-step generation**, we added a comparison between our method and one-step generation with post-processing in Section F (and Table A4).
- To **analyse the performance with different numbers of audio events**, we added an experiment on the VGGSounder dataset in Section H and Figure A4.
- To **demonstrate the generalizability of the proposed method across datasets**, we added experiments on the AudioCaps test set and the Movie Gen Audio Bench in Section I (and Tables A6 and A7).

Please see the corresponding part in the updated manuscript, as well as our responses to each review.

---

### Author Response · Authors · 2025-12-01

Dear Area Chairs and Reviewers,

We express our sincere gratitude to all the reviewers for their time and invaluable feedback.
We would also like to express our sincere appreciation to the Area Chairs for taking the time to engage in this unexpectedly laborious decision-making process.
Although we do not have an opportunity to receive further feedback from the reviewers, we would like to summarize our work and the efforts made during the rebuttal period.

We are encouraged that the reviewers acknowledged the **clarity and novelty of the problem formulation** (yhaW, RTJi, d7qG, mo18), the **conceptual simplicity and strong theoretical foundation of Negative Audio Guidance (NAG)** (RTJi, mo18), the **practical advantages for training on a single-reference dataset** (yhaW, mo18), the **compatibility with existing flow-matching or diffusion frameworks** (d7qG), and the **comprehensive and strong empirical evidence** (yhaW, RTJi, mo18) including both objective and subjective evaluations.

During the rebuttal period, we made every effort to address the reviewers' concerns.
The table below summarizes the additional experiments conducted in response to the feedback.

|Reviewer's concern|Our revision/response|
|---|---|
|Reliability of the user study.|Added **confidence intervals to the user study** to show statistical significance (**Tables A1 and A2 in Section A**).|
|Motivation for step-by-step generation over single-step generation with post-processing.|Added **comparison between step-by-step and single-step + post-processing** in **Section F and Table A4**.|
|Analysis of the number of generation steps.|Added evaluations and analysis using the **VGGSounder** in **Section H and Figure A4**.|
|Generalizability of the proposed method across datasets.|Added evaluations on **AudioCaps and Movie Gen Audio Bench** in **Section I and Tables A6 and A7**.|
|Significance of improvements over MMAudio.|Provided **standard deviation** for the results of Tables1 and 2 in our response to mo18.|

In all these additional experiments, our method **consistently demonstrated clear improvements**. We believe these results directly address the reviewers’ concerns and further validate the effectiveness and generality of the proposed approach.

In addition to these experiments, we provided detailed explanations to address other concerns.
We will summarize our responses to all weaknesses and questions in the next post, "Rebuttal summary".
Please also refer to the full responses to each reviewer for details.

Once again, we sincerely thank all reviewers for their thoughtful and valuable feedback.
We are also grateful to the Area Chairs for dedicating their time and effort to evaluating our manuscript.

Best regards,
Authors.

---

> ### Author Response · Authors · 2025-12-03
> **Rebuttal summary**
>
> ## Reviewer yhaW
> Four weaknesses (W1-4) and two questions (Q1-2):
>
> - **(W1/Q2)**: Step-by-step generation vs. one-step + AudioSep for multiple track generation.
>     - **Ans**: We added **direct comparison with MMAudio + AudioSep**, showing clear gains in **separability, quality, and audio-text/video alignment** (Section F and Table A4).
> - **(W2)**: VLM-based evaluation prompts without ground-truth audio.
>     - **Ans**: We clarify that **evaluation prompts need not match ground-truth**, consistent with Foley practice. Section C has been expanded to explain the limitations and rationale of VLM-based prompting.
> - **(W3/Q1)**: NAG vs. existing negative prompting.
>     - **Ans**: We positioned both under a generalized guidance framework. Novelty is in **sequential V2A usage**: (i) using **previous audio as a negative condition**, and (ii) enabling **sequential generation with single-reference training data**
> - **(W4)**: Goal of this paper and choice of composited audio evaluation.
>     - **Ans**: Our goal is **higher controllability** while preserving quality and alignment. Simple summation is chosen to avoid heuristics and expose track-level improvements transparently.
>
> ---
>
> ## Reviewer RTJi
> Four weaknesses (W1-4) and five questions (Q1-5):
>
> - **(W1/Q1)**: Does NAG actually suppress previous audio events?
>     - **Ans**: Yes. **CLAP A-A** (Table 2) is computed over all track pairs and shows consistent improvement, confirming effective suppression. Our demo also supports this (the 5th track vs. others).
>
> - **(W2/Q2)**: Generatizability across datasets.
>     - **Ans**: We added evaluations on **AudioCaps** and **Movie Gen Audio Bench**, where our method shows consistent gains (Section I and Tables A6-A7).
>
> - **(Q3)**: Why use Qwen2.5-VL?
>     - **Ans**: Preliminary tests compared Qwen2.5-VL, LLaVA, and PPLLaVA; Qwen2.5-VL offered the best caption quality and accessibility.
>
> - **(W3/Q4)**: Applicability to other V2A architecture.
>     - **Ans**: Other models **lack conditioning mechanisms** or are **closed-source**, preventing fair comparision. Only MMAudio satisfies the requirements.
>
> - **(W4/Q5)**: Reliability of the user study.
>     - **Ans**: We added **95% confidence intervals** for both composited-audio preferences and per-track rating (Table A1-A2), confirming statistical significance.
>
> ---
>
> ## Reviewer d7qG
> Two weaknesses (W1-2) and four questions (Q1-4):
>
> - **(W1/Q2)**: Performance dependence on the number of audio events.
>     - **Ans**: We added **VGGSounder experiments** (Section H and Figure A4). Our method shows consistent gains in separability without quality loss, with larger benefits for more events.
>
> - **(W2/Q3)**: Contribution relative to MMAudio.
>     - **Ans**: We provided comprehensive **quantitative and qualitative evaluations** on both **individual tracks and composited audio**, illustrating a meaningful gain in multiple aspects.
>
> - **(Q1)**: Improvement on temporal alignment.
>     - **Ans**: This work focuses on **separability** via NAG, not further temporal alignment.
>
> - **(Q4)**: Relation to ReWaS and SonicVisionLM.
>     - **Ans**: Section 2 has been expanded to discuss both works.
>
> ---
>
> ## Reviewer mo18
> Two weaknesses (W1-2) and three questions (Q1-3):
>
> - **(W1)**: Dependence on base model.
>     - **Ans**: Our method can be applied to any diffusion/flow-matching model and offers **orthogonal controllability gains** independent of backbone improvements.
>
> - **(W2)**: Sensitivity to generation order.
>     - **Ans**: This reflects **evaluation constraints**, not practical limitations. ImageBind order ensures reproducibility; users naturally decide order in the real workflow.
>
> - **(Q1)**: Standard deviations for Tables 1-2.
>     - **Ans**: We provided **std across five runs**, and the results are consistent and meaningful.
>
> - **(Q2)**: Alternative mixing strategies
>     - **Ans**: Not explored. Our focus is on well-separated tracks rather than mixing strategies.
>
> - **(Q3)**: Inconsistency between Tables 1, 2, and A3.
>     -  **Ans**: Table A3 uses Random Order (Table A5). Tables 1-2 use **ImageBind order**. No inconsistency exists.

---

### Meta-Review · Area_Chair_QisS · 2026-01-10

**Summary:**

One concern from the reviewer is that the number of participants is limited. The reviewers also have questions about the motivation for step-by-step generation over single-step generation with post-processing, the reliance on the base model MMAudio, the generalizability of the proposed method on other datasets, and marginal improvements over MMAudio. Also, a reviewer expresses that the simple summation used for mixing audio is a critical point of failure for realism. The improvement over MMAudio is limited. The proposed method is affected by the generation order.

**Reviewer Concerns:**

The reviewers have provided an author response to the reviewers' concerns. Most questions are addressed by the reviewers. However, the AC believes some concerns are still outstanding. The success of the proposed method is heavily reliant on the underlying video-to-audio model MMAudio, as its performance is tied to MMAudio. Also, the gain over the MMAudio baseline or its variant is marginal. Also, the performance is sensitive to the generation order. While the authors believe that users will determine the order, there is no deeper anaylsis what generation order would be optimal. Given these limitations, the AC does not recommend acceptance to ICLR.

**Reviewer Scores:**

This paper has scores from reviewers: 2,4,6, and 6. Some reviewers may increase their scores if they particiate fully in the discussion.

---

### Decision · Program_Chairs · 2026-01-26

Reject